



# Characteristics of convective boundary layer and associated entrainment zone as observed by a ground-based polarization lidar

Fuchao Liu[1,2,3], Fan Yi[1,2,3], Zhenping Yin[1,2,3], Yunpeng Zhang[1,2,3], Yun He[1,2,3], Yang Yi[1,2,3]

[1]School of Electronic Information, Wuhan University, Wuhan, 430072, China

[2]Key Laboratory of Geospace Environment and Geodesy, Ministry of Education, Wuhan, 430072, China

[3]State Observatory for Atmospheric Remote Sensing, Wuhan 430072, China

*Correspondence to*: Fuchao Liu (lfc@whu.edu.cn), Fan Yi (yf@whu.edu.cn)

**Abstract**

A tilted polarization lidar (TPL) with a pointing angle of 30 ° off zenith has been developed for continuous monitoring of the atmosphere with 10-s time and 6.5-m height resolution. From lidar-derived aerosol backscatter, instantaneous ABL depths are retrieved by logarithm gradient method (LGM) and Harr wavelet transform method (HWT), while hourly-mean ABL depths by variance method. A new FWHM method utilizing the full width at half maximum (FWHM) of the variance profile of aerosol backscatter ratio (ABR) fluctuations is proposed to determine the entrainment zone thickness (EZT). Both typical winter and summer clear-day observational cases are presented. It is concluded the convective boundary layer (CBL) evolution can be described by four stages. At the formation stage, the hourly-mean CBL depth grew slowly with a positive growth rate of <0.15 km/h. At the growth stage, the hourly-mean CBL depth grew fast with average growth rate of >0.3 km/h. At the quasi-stationary stage, the hourly-mean CBL depth varied little and the corresponding growth rate changed sign with absolute value of <0.15 km/h. At the decay stage, the hourly-mean CBL depth kept decreasing until the layer being re-categorized as a residual layer. The instantaneous CBL depths exhibited different fluctuation magnitudes in the four stages and fluctuations at the growth stage were generally more obvious. The EZT is investigated by the FWHM method. It is found that for the same statistical time interval of 0900-1900 LT, the winter case had smaller mean (*mean*) and standard deviation (*stddev*) of EZT data (a *mean* of 94 m, a *stddev* of 38 m) than those of the summer case (a *mean* of 127 m, a *stddev* of 49 m); besides, the former had respective percentages of 8.5% and 7.5 % of EZT falling into the subranges of 0-50 m and >150 m, while the latter had respective percentages of 2.0% and 31 % of EZT falling into the same corresponding subranges. Common statistical characteristics also existed for both cases. The growth stage always had the largest *mean* and *stddev* of EZT and the quasi-stationary stage usually the smallest *stddev* of EZT. For all four stages, most EZT values fell into the 50-150 m subrange; the overall percentages of EZT falling into the 50-150 m subrange between 0900 and 1900 LT were 84% and 67% for the winter and summer cases, respectively.



## 1 Introduction

Monitoring the atmospheric boundary layer (ABL) is of essential importance since the ABL is in direct contact with nearly all terrestrial life on earth (Lammert et al., 2006). The ABL locates at the lower part of the troposphere and subjects to influences of various processes. These processes, including land or water surface exchanges at the bottom and entrainments at the top, govern the transport of heat, momentum, moisture and substances (e.g., aerosols and other constituents) between the ground and the free atmosphere (FA) (Stull, 1988; Pal et al., 2010).

The depth (or height) of the ABL is a key parameter for parameterization of the ABL, as it determines the available volume for pollutants dispersion and resulting concentrations (Pal et al., 2015; Li et al., 2017; Su et al., 2018; Su et al., 2020), as well as the region dimension in which transport processes can take place. The ABL depth is defined as the interfacial height that separates the ABL and the FA (Stull, 1988). It actually exhibits apparent diurnal evolution following the local surface temperature variation with a magnitude from a few tens of meters to several kilometers (Kong and Yi, 2015). In clear daytime after sunrise, the ABL depth generally increases first as convective activities intensify, then decreases after reaching its maximum in the afternoon when turbulence intensity decays. The convectively-driven ABL is designated as convective boundary layer (CBL). After sunset, the CBL is replaced by stable boundary layer (SBL; or nocturnal boundary layer, NBL) with a much lower depth. Because the convective processes driven by the sensible heat flux at the surface can be reflected by tracer (e.g., water vapor and aerosols) concentration within the CBL and in various atmospheric variables, multiple methods based on tracers and distinct instrumentations have been utilized to determine the CBL depth (Behrendt et al., 2011a; Cimini et al., 2013; Sawyer and Li, 2013). In-situ radiosonde measurement serves as one popular way to derive CBL depth (Seidel et al., 2010; Guo et al., 2019) for its wide distribution all over the world and long observation history which makes it suitable for CBL depth climatology study (Dang et al., 2019) despite of its low temporal resolution (usually 2–4 times per day). From radiosonde profiles of temperature, pressure, humidity and wind, the CBL depth can be retrieved by parcel method (Hennemuth and Lammert, 2006; Seidel et al., 2010), Richardson method (Seibert et al., 2000; Seidel et al., 2010; Zhang et al., 2013), and gradient method (Seidel et al., 2010). Ground-based remote sensing instruments, such as sodar (Helmis et al., 2012), microwave radiometer (Cimini et al., 2013), wind profiling radar (Liu et al., 2019), ceilometer (Zhu, 2018) and lidar, favour continuous monitoring of the CBL depth at a fixed location; space-borne lidar like Cloud-Aerosol Lidar with Orthogonal Polarization (CALIOP), on the other hand, can provide global coverage, but suffers from low signal-noise ratio (SNR) at daytime for CBL measurements (Liu et al., 2015; Zhang et al., 2016; Su et al., 2017). Among these remote sensing techniques, lidar can continuously measure the atmospheric backscatter with high spatial and temporal resolution which thus enables detailed study on the microscale structures in the CBL. Based on the lidar-derived backscatter information from given trace substances (e.g., water vapor and aerosols), the ABL depth can be determined either by process-based variance method (e.g., Lammert et al., 2006; Martucci et al., 2007; Wulfmeyer et al., 2010; Pal et al., 2013; Kong and Yi, 2015), or by vertical-distribution-based method (e.g., the derivative method; the Harr wavelet transform method) (Cohn et al., 2000;



Brooks, 2003; Morille et al., 2007; Baars et al., 2008; Pal et al., 2010; Granados-Muñoz et al., 2012; Lewis et al., 2013;
Sawyer and Li, 2013; Su et al., 2020).

Turbulence is a frequent phenomenon in the CBL and turbulent mixing serves as an effective mechanism resulting in
homogeneous distribution of scalars (e.g., humidity, aerosols and other constituents) in middle and lower parts of the CBL
(Manninen et al., 2018). The middle and lower parts of the CBL characterized by evenly mixing is also called mixing layer
(ML). However, near the top area of the CBL, sharp gradient of scalars might appear due to vigorous mixing of overshooting
thermals (by updraft) and FA air (by downdraft) (Stull, 1988). This region corresponds to the entrainment zone (EZ).
Entrainment processes occurred in the EZ controls the CBL growth and structure, as well as clouds formation and
distribution in the CBL (Brooks et al., 2007). Entrainment rate is an important parameter for understanding the fundamental
physical entrainment processes; however, this parameter cannot be directly measured but needs to be inferred from other
measurement results (Lenschow et al., 1999). The entrainment zone thickness (EZT) provides a possible approach for
parameterizing the entrainment rate (Deardorff et al., 1980). The top of EZ can be regarded as the highest height that the
thermal within a region reaches (Stull, 1988), while the bottom of EZ is difficult to define and usually taken subjectively as
the height where about 5-10% of the air on a horizontal plane has the FA characteristics (e.g., Deardorff et al., 1980; Wilde
et al., 1985). The EZT is determined by the top and bottom heights of the EZ and measures the averaged vertical size of the
ABL-height fluctuation (Boers et al., 1995). Since small scale processes often become important in the EZ due to high
variability of the scalar distribution in these regions, determination of EZT requires monitoring of tracers with very high
temporal-spatial resolution in this area. Based on lidar-retrieved high-resolution time series of instantaneous ABL depth, the
standard deviation technique (e.g., Davis et al., 1997) and the cumulative frequency distribution method (e.g., Wilde et al.,
1985; Flamant et al., 1997; Pal et al., 2010) have been employed to investigate the EZT.

In this work we present the measurement results of the CBL and associated EZ using a recently-developed titled polarization
lidar (TPL). The TPL is housed in a specially-customized working container and capable of operating under various weather
conditions (including heavy precipitation). The TPL has an inclined working angle of 30 ° off zenith and routinely monitors
the atmosphere with a time resolution of 10 s and a height resolution of 6.5 m. The equivalent minimum height with full
overlap for the TPL is ~173 m above ground level (AGL). Thus this TPL provides a possibility to perform detailed study on
the ABL. The instrument, methodology, observational results and summary and conclusions are stated successively in
following sections.
**2 Instrument**
The TPL locates in the campus of Wuhan University, Wuhan, China (30.5 °N, 114.4 °E and 70 m above sea level). Figure 1a
shows a schematic optical layout of the lidar system. The lidar transmitter introduces a solid Nd:YAG laser to generate an





emission of 70 mJ per pulse at 532 nm with a repetition of 20 Hz. A Brewster polarizer (PR) improves the linear polarization
purity of the outgoing laser light before entering the beam expander (BE). The $3\times$BE compresses the divergence of the laser
to be <0.25 mrad. A steerable reflecting mirror (RM) then guides the expanded beam into atmosphere. In the receiver, a
Cassergrain telescope collets the atmospheric backscatter. The telescope has a clear aperture of 203.2 mm and a focal length
of 2032 mm. The subsequent optics contains an iris, a collimating lens (CL), a half-wavelength plate (HWP), a RM and an
interference filter (IF). The iris sets the telescope field of view to be 1.0 mrad. The HWP guarantees the polarization plane of
the propagating light beam to be exactly coincident with the receiver polarization analyzer. The IF has a bandwidth of 0.17
nm centered at 532 nm and a peak transmittance of 79%. After being filtered by the IF, the parallel and perpendicular
polarization light components are detected by two detection channels (designated as the P- and S-channel, respectively). In
each of the P- and S-channel, two cubic polarization beam splitters (PBS) are cascaded to reduce crosstalk between the two
orthogonal polarization channels; a focusing lens (FL) then focuses the signal light on the photosensitive surface of
subsequent photomultiplier tube (PMT); neutral density filters (not shown here) are also added before the FL to avoid
saturation of the PMT. Finally, a PC-controlled two-channel transient digitizer (TR20-160, Licel) records the detected
signals as raw saved data with a time resolution of 10 s and range resolution of 7.5 m.

Figure 1b provides a picture of the TPL transmitting-receiving optics. The whole optics is installed on a mechanical tilted
platform (TPF) with a fixed elevation angle of 30 °. This translates a same angle of the telescope optical axis off zenith.
Besides, the TPL system is housed in a specially-customized working container with temperature and humidity control. The
working container opens a window on one side that permits the propagating laser beam and atmospheric backscatter to pass
through without blocking. The working container enables the TPL to operate under various weather conditions including
heavy precipitation.

The whole transmitting-receiving optics of the TPL has a compact arrangement and the tested minimum range with full
overlap is 200 m. Given the 30 ° tilted angle off zenith, this yields an equivalent height of ~173 m AGL. Thus the TPL partly
provides a possibility of the depth investigation of shallow CBL and NBL. The channel gain ratio of the TPL was calibrated
after its foundation using sky background method (Wang et al., 2009). Specifically, the calibration was performed when the
sky was clouded over so that the background sun light could be regarded as totally unpolarized. The gain ratio turned out to
be 0.09521$\pm$0.00031. It is further investigated that the lidar-measured molecular volume depolarization $\delta_{V,m}$ in clear areas is
0.00780$\pm$0.00072. Considering the theoretical $\delta_{V,m}$ for this TPL should be 0.00364 (Behrendt et al., 2002), the offset value of
0.00416 due to depolarization effect of the lidar system is rather small and thus neglected.



## 3 Methodology

### 3.1 Method to determine ABL depth

The Licel-recorded raw analog and photon count data are first used to generate a reasonable photon count profile with larger dynamic range based on a developed gluing algorithm (Newsom et al., 2009; Zhang et al., 2014). This glued photon count profile remains a temporal resolution of 10 s and a range resolution of 7.5 m. Simultaneous the obtained P- and S-channel signals, the unpolarized range-square corrected elastic signal $X$ at range $R$ can be reconstructed by:

$$X(R) = [N_p(R) + GR \cdot N_s(R)] \cdot R^2 \tag{1}$$

where subscripts $p$ and $s$ denote P- and S-channel, respectively. $N$ is the background-subtracted photon count signal. The channel gain ratio $GR$ has already been determined as stated before.

Since the TPL is slantingly-pointed with an angle of 30 ° off zenith, the range $R$ can be readily converted to corresponding height $z$ by multiplying a factor of cos30 °. Hereafter in this work we use height $z$ instead of range $R$. From the range-square corrected elastic signal $X$, the vertical-distribution-based method can be employed to determine an ABL depth for each $X$ profile. Here both the logarithm gradient method (LGM) (e.g., Wulfmeyer, 1999; Pal et al., 2010) and Harr wavelet transform method (HWT) (e.g. Davis et al., 2000; Brooks, 2003) are tested to retrieve ABL depth.

The ABL depth $z_{LGM}$ determined by LGM method is defined as:

$$z_{LGM} = \min[D(z)] = \min\left[\frac{dlnX(z)}{dz}\right] \tag{2}$$

where $D$ stands for the derivative of logarithmic $X$.

The ABL depth $z_{HWT}$ determined by HWT method is defined as:

$$
\begin{aligned}
z_{HWT} &= \max[W_f(a,b)] = \max\left[\frac{1}{a}\int_{z_{min}}^{z_{max}} X(z)H\left(\frac{z-b}{a}\right)dz\right] \\
&\quad for\ z_{min} < b < z_{max} \\
&and\ H\left(\frac{z-b}{a}\right) = \begin{cases} 1, & b-a/2 \leq z \leq b \\ -1, & b < z \leq b+a/2 \\ 0, & elsewhere \end{cases}
\end{aligned}
\tag{3}
$$





in which $W_f$ is the covariance transform value, $H$ the Harr wavelet function. The dilation $a$ is tested and set to be 200 m for
this work. $z_{min}$ and $z_{max}$ are the lower and upper heights for the lidar signal profile, respectively.

The advantage of applying the LGM and HWT methods is that an instantaneous ABL depth can be determined according to
each $X$ profile which favors a high temporal resolution. However, in case of residual layer (RL) or multiple aerosol layers,
usually several local minima occur for the retrieved $D$ profile, making the choice of the true minimum for the LGM method
difficult (Menut et al., 1999; Pal at al., 2010). As for the HWT method, when the ABL is shallow (e.g., for the NBL and the
early stage of the CBL after sunrise), subjective constrain on the upper integral height $z_{max}$ needs to be made to the base of
existing aerosol layers aloft (Gan et al., 2011). All these situations hinder the LGM and HWT methods from an automated
and robust attribution of the ABL depth.

To find a more reliable method suitable for an automated procedure, the process-based variance method can be utilized to
provide a reference for the search of a local minimum by the LGM method, or the search of a local maximum by the HWT
method in a given time interval (e.g., Lammert et al., 2006; Pal et al., 2013). In this work the variance profile of aerosol
backscatter ratio (ABR) fluctuations is calculated and the height with maximum variance is assigned as ABL depth. Here the
ABR profile is retrieved using Fernald backward iteration method given a fixed lidar ratio (Fernald, 1984; Behrendt et al.,
2011b). The fixed lidar ratio is chosen to be 50 $sr$ at 532 nm according to existing measurement results of urban aerosols
(e.g., Ansmann et al., 2005; Müller et al., 2007). Typical time interval is 1 h for generating a variance profile. Note this
variance method determines a mean ABL depth for the given 1-h time interval. To attribute the instantaneous ABL depth in
the same time interval, the height with local minimum/maximum by the LGM/HWT method nearest to the hourly mean ABL
depth by the variance method is selected.

The remaining problem is that several local peaks might also appear for the variance profile in case of multiple (residual)
aerosol layers. This problem is settled by visualizing the contour plots of $D(z)$ and $W_f(z)$ to limit a proper height range for
variance calculating. As an example, Figure 2 shows the calculated $D(z)$ and $W_f(z)$ in the height range of 0-2.5 km on Jan 31,
2020. Sunrise (SR) and sunset (SS) times are marked by thick black dashed lines. As seen in Figure 2, before 1000 local time
(LT) multiple (residual) aerosol layers above 0.5 km were clearly indicated by stripes of local minima of $D(z)$ and maxima
of $W_f(z)$; besides, advected aerosols above 0.7 km were also discernible after 1930 LT (see also in Figure 4). From Figure 2,
it is found that an abundant aerosol layer subsided from around 1.25 km at 0000 LT to about 0.6 km at 1000 LT. This layer
definitely leads to misattribution of ABL depth by the automated procedure using the LGM and HWT methods, as well as
that by the variance method. By visualizing these contour plots, it is intuitive and convenient to distinguish and locate the
above misguiding aerosol layers. Then proper upper height limits for applying the variance method can be correctly
determined as the real ABL should be below these multiple (residual) aerosol layers aloft. Around 1930 LT after SS, the





subsided CBL near 0.6 km should be re-categorized as a RL. Again, the proper upper height limits for applying the variance
method shall be set below the RL for the ABL (NBL) depth determination after 1930 LT.

**3.2 Method to determine EZT**

Since simultaneous measuring of the atmosphere in a large horizontal plane is actually difficult, an equivalent continuous
sampling in the time domain at a fixed monitoring site is favored and can be easily performed, given the Taylor's hypothesis
of "frozen turbulence" theory (Stull, 1988). Under this assumption and from the retrieved time series of instantaneous ABL
depth, the standard deviation technique (e.g., Davis et al., 1997) and the cumulative frequency distribution method (e.g.,
Wilde et al., 1985; Flamant et al., 1997; Pal et al., 2010) can be employed to obtain the EZT. However, the values of EZT
obtained by these two methods exhibit obvious discrepancies (e.g., Pal et al., 2010). The choice of specific percentage of air
having the FA characteristics for the definition of EZ bottom height is rather subjective and seems variable between different
researchers. Moreover, considering that variations of ABL depths can result from not only entrainment but also non-
turbulent processes (e.g., atmospheric gravity waves and mesoscale variations in ABL structure), the above methods might
not really characterize the true EZ (Davis et al., 1997). This situation motivates us to develop a new approach to determine
the EZT in this work.

Let's revisit the definitions of the top and bottom heights of the EZ firstly given by Deardoff et al. (1980) and Wilde et al.
(1985) that have respectively 100% and 5-10% of air on a horizontal plane sharing the FA characteristics. It's concluded the
top and bottom heights, especially the bottom one, are defined in a statistically averaging manner. Besides, when observed
from a perspective of physical process, entrainment mixing of clean FA air and well mixed ML air generally results in
significant fluctuations of scalars (e.g., number density of aerosols) in the EZ (see later in Figure 4 and Figure 7). In the
absence of clouds and advected aerosols, the fluctuation magnitudes of aerosol number density in the EZ are usually larger
than those in the FA and ML. Taking all above into consideration, the variance of ABR fluctuations is utilized here to
statistically represent the fluctuations of aerosol number density. Subsequently the full width at half maximum (FWHM) of
the variance profile of ABR fluctuations can be employed to define the EZ, as this FWHM records the recent mixing history
and quantitatively indicates in which area the larger variations of aerosol number density (ABR) take place. In detail, the
height with maximal variance in a variance profile calculated in a given time interval is firstly located as the ABL depth; this
is coincident with the definition by the variance method. Then, the upper and lower heights with half value of the maximum
variance are searched and defined as the top and bottom heights of EZ, respectively. The EZT is consequently determined by
the height interval between the searched top and bottom heights of EZ. This method is designated as FWHM method here.

As an example, Figure 3 illustrates the FWHM method of using the variance of ABR fluctuations to determine the EZT. In
figure 3a, the profile of standard deviation of ABR, $\sigma$(ABR), is first calculated for a chosen time interval and plotted as thin
black line. From this $\sigma$(ABR) profile, the CBL depth (indicated by the dotted line) is definitely located at the height with





maximum $\sigma$(ABR). For a strong updraft (as is this case) that carries ML air upward into the FA, intense fluctuations occur in
the EZ while less-intense fluctuations in the ML and FA. Therefore the corresponding $\sigma$(ABR) profile exhibits much larger
values near the CBL depth, as well as clear-cut steep upper and lower edges on each side of the CBL depth. Then the
FWHM of the $\sigma$(ABR) profile can be directly and easily determined, which further defines the EZ as well as the
corresponding EZT (thick vertical line). However, Figure 3a only stands for an ideal situation, while real atmospheric
processes are usually much more complex. Figure 3b describes a less-intense updraft case that the lower edge of the $\sigma$(ABR)
profile is not clear-cut enough to locate the lower height of the EZ. In this situation, a quadratic polynomial fitting (dashed
line) is applied to the lower edge, so that the "contaminating" fluctuations in the ML is removed. Combining the upper edge
and the fitted lower edge, the true EZT is determined (thick vertical line). Note that only the clear-cut steep part of the lower
edge (nearly overlapping with the fitted line; see Figure 3b) is chosen for fitting and usually a quadratic polynomial function
exhibits satisfactory fitting performance. Figure 3c shows a case in the late afternoon when turbulence is decayed and
advected aerosols appear at higher heights. Consequently, neither the upper nor the lower edge of the $\sigma$(ABR) profile is
clear-cut enough. Then quadratic polynomial fittings (dashed lines) are applied to both edges to help determine the EZT
(thick vertical line). An automated procedure is hence developed to determine the EZT based on this FWHM method.
**4 Observational results**
In this section two typical ABL measurement results under clear weather conditions are presented. Note the TPL has an
equivalent minimum height of ~173 m with full overlap, the retrieved results (e.g., ABR) below 173 m shall not be
reasonable and discussions are confined only to heights above this value. Before making subsequent physical analysis on the
retrieved results, the corresponding conversion of range $R$ to height $z$ is valid under the assumption that the aerosols are
horizontally homogeneous in the related horizontal space. To state this issue, the ABR results by this TPL and another co-
located vertically-pointing 532-nm polarization lidar (Kong and Yi, 2015) in our lidar site were compared. The comparisons
showed that the concurrent ABR profiles by these two lidars always (at least in the ABL region) had nearly identical
structures and magnitudes. This convinced the above assumption and the conversion could be made straightforward. Besides,
here we focus mainly on the CBL in this work.
**4.1 Case study 1 (Jan 31, 2020)**
Figure 4 presents a full-day measurement result of the ABL performed in late winter. Figure 4a provides a time-height
contour plot (10-s time and 6.5-m height resolution) of ABR on Jan 31, 2020. It is seen that the atmosphere was quite clear
in height ranges between 1.7 and 2.5 km, while multiple (residual) aerosol layers were present below 1.7 km until 1400 LT
when they were totally "engulfed" by the well-developed CBL. Advected aerosol layers above ~0.6 km were also discernible
after 1930 LT. In spite of the presence of these aerosol layers aloft, the variance method is first applied to retrieve the hourly
mean ABL depth for each 1-h time interval. Before finding a local maximum from the calculated ABR-variance profile, the





proper upper and lower height limits are determined by visualizing the corresponding $D(z)$ and $W_f(z)$ contour plots (see
Figure 2). Then the height with local maximal variance between the chosen upper and lower heights is searched and located
as the ABL depth (red solid circles). SR and SS times are indicated by thick black dashed lines. As shown by Figure 4a, the
values of ABR in the CBL had a direct "response" to the development of CBL depth: between ~1030 and 1130 LT when the
initial CBL was shallow (CBL depth <0.35 km), the ABR had larger values reaching 10; then as the CBL depth increased
and reached to a maximum of ~1.02 km around 1330 LT, the ABR values in the CBL generally decreased. If we assume that
in the lidar-observation time interval the probed aerosols didn't undergo chemical and physical reactions, then the change in
ABR values can be regarded to the change of aerosol number density in the CBL (Engelmann et al., 2007; Pal et al., 2010).
Figure 4a graphically describes the vertical transport of aerosols from surface to upper heights: as the available dispersion
volume (CBL depth) enlarges, the ABR values (the mixed aerosol number density) fall. Between 1330 and 1830 LT, the
ABR values in the CBL exhibited features of vertical homogeneity (see Figure 4b), indicating the fully mixing of aerosols in
the ML.
Figure 4b over-plots the ABR profiles (thin black lines) in each 1-h time interval. The hourly mean ABR profile is also
added (blue line). It is found that the fluctuation features of the over-plotted ABR profiles differ at distinct developing stages
of the CBL. In the time interval between ~0830 and 1130 LT, the hourly mean CBL depth grew slowly from ~0.18 km at
around 0830 LT to ~0.35 km at around 1130 LT; meanwhile, fluctuations of the over-plotted ABR profiles increased in this
initial CBL. This stage corresponds to the formation period of the CBL. After SR, the sun started to heat the surface.
Consequently convective activities started to occur and CBL began to develop, but the CBL depth growth was restricted by
the upper stable NBL (Stull, 1988). Then the hourly mean CBL depth increased rapidly from ~0.35 km at around 1130 LT to
~1.02 km at around 1330 LT; fluctuations of the over-plotted ABR profiles kept increasing at first throughout the CBL, then
decreased and tended to become uniform in the middle and lower parts of the CBL. This stage denotes the rapid growth
period of the CBL. After ~1130 LT the cool NBL air was warmed to a temperature near that of the above RL, and the CBL
top had reached the base of the RL. At this point the stable NBL capping the CBL vanished, so that thermals could penetrate
upward quickly, allowing the growth of the CBL depth with a larger growth rate. However, this rapid growth did not
continue after the CBL depth reached the top of the RL, where the FA above prevented thermals from further vertical motion
(Stull, 1988). Accompanying the initial penetrating thermals upward, aerosols (as well as other constituents) were
transported vertically and turbulently mixed, exhibiting a high fluctuation feature for the ABR in the CBL; while as vertical
transport and turbulent mixing continued, aerosols shall be fully mixed in a larger available volume, reflected by both
smaller fluctuations of the ABR profiles and values of ABR themself. Next, the hourly mean CBL depth changed very little
from ~1.02 km at around 1330 LT to ~0.96 km at around 1630 LT; fluctuations of the over-plotted ABR profiles kept
decreasing until all the ABR profiles became uniformly upright below the top area of the CBL. This stage represents the
quasi-stationary period of the CBL. The little change of the CBL depth is governed by the balance between entrainment and
subsidence (Stull, 1988). In this stage, the aerosols had been fully and evenly mixed in the ML, indicated by the smallest



288 fluctuations of the ABR profiles and values of ABR. Finally in the late afternoon, the hourly mean ABL depth maintained

289 decreasing from ~0.96 km at around 1630 LT to ~0.39 km at around 1930 LT; fluctuations of the over-plotted ABR profiles

290 increased slightly in the ML. This stage describes the decay period of the CBL. As the solar radiation weakened, the strength

291 of convective turbulence reduced so that turbulence could not be maintained against dissipation (Nieuwstadt et al., 1986).

292 The small increase in ABR fluctuations reflected that the decay turbulence could no longer preserve the homogeneous

293 distributions of the aerosols in the ML. After SS the turbulence in the ML might decay completely, then the layer needed to

294 be re-categorized as a RL while at the same time NBL had already formed near surface. It should be noted that for all the

295 four stages, obvious fluctuations of the over-plotted ABR profiles were always present near the top area of the CBL. This

296 fluctuating behavior looked like a "node", representing the structure of the EZ between the CBL and FA (Kong and Yi,

297 2015).

299 Figure 5 investigates further the evolution of the CBL depth on Jan 31, 2020. Figure 5a plots the instantaneous CBL depths

300 (blue) obtained by LGM method (before 1000 and after 1900 LT) and HWT method (between 1000 and 1900 LT). For

301 comparison, the hourly mean ABL depths (red solid circles) by variance method are added. Figure 5b shows the

302 corresponding hourly mean ABL depth growth rate. At the formation stage, the CBL depth growth rate changed sign from

303 negative to positive at ~0830 LT and reached a maximum of ~0.084 km/h at around 1000 LT. After SR, the ABL depth did

304 not increase immediately until later (the growth rate be negative before ~0830 LT). The time interval between SR and 1130

305 LT is roughly defined as the early morning transition (EMT) period (Pal et al., 2010). During this EMT period, the

306 instantaneous CBL depth generally exhibited small deviation from that indicated by the hourly mean ABL depth (red line).

307 At the growth stage, the CBL depth increased with a mean growth rate of > 0.3 km/h and a maximum growth rate of ~0.36

308 km/h at around 1200 LT. Meanwhile, the instantaneous CBL depths showed obvious larger deviations and fluctuations. At

309 the quasi-stationary stage, the CBL depth growth rate changed sign at around 1430 LT and varied between 0.09 and -0.12

310 km/h. The accompanying instantaneous CBL depths had comparatively moderate deviations and fluctuations. At the final

311 decay stage, the ABL depth growth rate kept negative with a minimum of -0.40 km/h at around 1900 LT. The fluctuations of

312 instantaneous CBL depth were generally moderate before SS. The ABL depth growth rate returned to nearly zero at ~2000

313 LT and the time interval between SS and 2000 LT is roughly defined as the early evening transition (EET) period (Pal et al.,

314 2010). During this EET period, the instantaneous ABL depth exhibited small deviation from that indicated by the hourly

315 mean ABL depth (red line).

317 It is visually observable that the time series of instantaneous CBL depth fluctuate on small time scales (Figure 5a), especially

318 in the growth stage, reflecting the entrainment characteristics in the EZ. As already mentioned before, the EZT serves as a

319 measure of averaged vertical size of the ABL-depth fluctuation (Boers et al., 1995). Hence the EZT is calculated and

320 investigated here. Figure 6a plots the CBL depth $Z\_CBL$ (red) obtained by the variance method between 0900 and 1900 LT

321 on Jan 31, 2020. The EZ upper height $Z\_Upper$ (magenta) and lower height $Z\_Lower$ (blue) are determined from the FWHM





of the $\sigma$(ABR) profile (see Figure 3). To generate one $\sigma$(ABR) profile, a group of 18 consecutive ABR profiles in a time
interval of 3 min is utilized. So that the retrieved $Z\_CBL$ and EZT represent the corresponding mean values in each given
time interval of 3 min. Here the choice of 3 min is a compromise between time resolution of EZT and reliability of $\sigma$(ABR)
profile. Figure 6b exhibits the resulting EZT (red) and ratio of EZT to $Z\_CBL$ (blue; for convenience, the ratio is multiplied
by a factor of 0.5 so that the two vertical axes share the same scaling range). The overall EZT time series between 0900 and
1900 LT had a minimum (*min*) of 26 m, a maximum (*max*) of 267 m and a mean (*mean*) of 94 m with a standard deviation
(*stddev*) of 38 m. The ratio values spanned a range from 3.5% to 76.8%. Larger ratio values (>30%) mainly appeared in the
formation stage and first half of growth stage of the CBL (before 1230 LT), while most ratio values were <20% after the
second half of the growth stage (after 1230 LT).

Table 1 summarizes the corresponding statistical data for all the four developing stages of the CBL on Jan 31, 2020. It is
found that the growth stage had largest EZT statistical data (a *min* of 65 m, a *max* of 267 m, a *mean* of 122 m and a *stddev* of
41 m). On the contrary, the quasi-stationary stage exhibited lower EZT statistical data (a *max* of 154 m, a *mean* of 82 m and
a *stddev* of 28 m except for a *min* of 39 m). The formation stage (a *min* of 33 m, a *max* of 158 m, a *mean* of 85 m and a
*stddev* of 36 m) and decay stage (a *min* of 26 m, a *max* of 180 m, a *mean* of 95 m and a *stddev* of 36m) generally showed
comparable statistics of EZT. When the values of EZT are divided into five subranges (see Table 1 for detail), it is observed
that the formation stage had a highest percentage of 16.0% of EZT falling into the 0-50 m subrange, while the growth stage
had none falling into the same subrange. However, the growth stage had the largest percentage of 17.5% of EZT falling into
the 150-200 m subrange, and was the unique stage having EZT value exceeding 200 m. The quasi-stationary stage had the
smallest percentage of 1.7% of EZT falling into the 150-200 m subrange. For all four stages, the EZT values mostly fell into
the 50-100 m and 100-150 m subranges with corresponding cumulative percentages of 80.0%, 80.0%, 88.3% and 86.0%,
respectively.
**4.2 Case study 2 (May 19, 2020)**
Figure 7 presents a full-day measurement result of the ABL executed in early summer. Figure 7a provides the time-height
contour plot (10 s and 6.5 m resolution) of ABR on May 19, 2020. On this summer day, there were less abundant aerosols
above 0.6 km compared to that below 0.6 km between 0000 and 1200 LT. Another advected aerosol layer starting at around
0900 LT (not indicated here) above 1.5 km subsided but did not interfere with the lower ABL. The variance method is first
used to determine the hourly mean ABL depth for each 1-h time interval (red solid circle). The ABR before 1030 LT showed
large values (>8) in the initial CBL below 0.4 km. Then as the CBL depth (red line) increased and reached to a maximal of
~1.15 km at around 1430 LT, the ABR values in the CBL exhibited a decrease below 0.4 km while an general increase
between 0.4 km and 1.0 km, indicating the turbulent transport of aerosols from surface to upper heights. Figure 7b over-plots
the ABR profiles (thin black lines) in each 1-h time interval and the hourly mean ABR profile (blue line). In the formation
period of the CBL, the hourly mean CBL depth grew slowly from ~0.18 km at around 0830 LT to ~0.56 km at around 1230





LT; fluctuations of the over-plotted ABR profiles prevailed throughout the CBL. Then in the growth period of the CBL, the
hourly mean CBL depth increased rapidly from ~0.56 km at around 1230 LT to ~1.63 km at around 1430 LT; observable
fluctuations of the over-plotted ABR profiles continued, but tended to decrease and become uniform in the middle part of
CBL. Next in the quasi-stationary period of the CBL, the hourly mean CBL depth changed very little from ~1.63 km at
around 1430 LT to ~1.52 km at around 1630 LT; fluctuations of the over-plotted ABR profiles decreased slightly and all the
ABR profiles became uniformly upright in the middle part of the CBL. Finally in the decay period of the CBL, the hourly
mean ABL depth kept decreasing from ~1.52 km at around 1630 LT to ~0.24 km at around 2030 LT; both fluctuations of the
over-plotted ABR profiles and ABR values exhibited small decrease in the middle and lower part of the CBL. Again for all
the four periods, obvious fluctuations of the over-plotted ABR profiles were always present near the top area of the CBL.

Figure 8a plots the instantaneous CBL depth (blue) obtained by LGM method (before 0900 and after 2000 LT) and HWT
method (between 0900 and 2000 LT). The hourly mean ABL depths (red solid circles) by variance method are added. Figure
8b shows the hourly mean ABL depth growth rate (red solid circles). At the formation stage, the CBL depth growth rate
changed sign from negative to positive at ~0800 LT and reached a maximal of ~0.14 km/h at around 0900 LT. The EMT
period is roughly defined between SR and 1200 LT. The instantaneous CBL depths exhibited small deviation from that
indicated by the hourly mean ABL depth (red line) before 1000 LT, but showed increased deviation later on. At the growth
stage, the CBL depth increased with a mean growth rate of > 0.48 km/h and a maximum growth rate of ~0.59 km/h at around
1300 LT; meanwhile, the deviations and fluctuations of the instantaneous CBL depths obviously enlarged. At the quasi-
stationary stage, the CBL depth growth rate changed sign to be negative at around 1500 LT and varied between -0.04 and -
0.07 km/h; the fluctuations of the instantaneous CBL depth remained obvious. At the final decay stage, the ABL depth
growth rate kept negative with a minimum of -0.58 km/h at around 2000 LT; the fluctuations of instantaneous ABL depth
were still observable. The ABL depth growth rate returned to nearly zero at ~2100 LT and the time interval between SS and
2100 LT is roughly defined as the EET period. During the EET period, the instantaneous ABL depth generally exhibited
small deviation from that indicated by the hourly mean ABL depth (red line). Note that after SS the CBL should be re-
categorized as a RL.

Figure 9a plots the CBL depth $Z\_CBL$ (red) obtained by the variance method between 0900 and 1900 LT on May 19, 2020,
as well as the EZ upper height $Z\_Upper$ (magenta) and lower height $Z\_Lower$ (blue) derived from the FWHM of the $\sigma$(ABR)
profile. Figure 9b shows the resulting EZT (red) and ratio of EZT to $Z\_CBL$ (blue). The overall EZT time series between
0900 and 1900 LT had a min of 42 m, a max of 331 m and a mean of 127 m with a stddev of 49 m. The ratio values varied
between 4.2% and 66.2%. Larger ratio values (>30%) mainly occurred in the formation stage and the initial of growth stage
of the CBL (before 1315 LT), while most ratio values were <20% later on (after 1315 LT).





Table 2 concludes the corresponding statistics for all the four developing stages of the CBL on May 19, 2020. It can be seen
that the growth stage had the largest *mean* (153 m) of EZT, while the formation stage exhibited the lowest *mean* (106 m) of
EZT. Besides, the growth stage and quasi-stationary stage had the largest *stddev* (57 m) and the smallest *stddev* (35 m) of
EZT, respectively. When the values of EZT are divided into five subranges (see Table 2 for detail), it is found that the
formation stage had a percentage of 5.7% of EZT falling into the 0-50 m subrange, while the other three stages had none
falling into the same subrange. For this summer case, all four stages had percentages of >15% of EZT falling into the 150-
200 m subrange, and the growth stage exhibited the largest percentage of 20.0% of EZT exceeding 200 m. Again for all four
stages, the EZT had values mostly falling into the range between 50 and 150 m with corresponding percentages of 75.7%,
52.5%, 75% and 60.0%, respectively.

Table 3 compares the EZT statistics for the winter and summer cases. As shown in Table 3, the two cases exhibited apparent
statistical differences. For the same time interval of 0900-1900 LT, the winter case (case 1) had overall statistical EZT data
(a *mean* of 94 m, a *stddev* of 38 m) smaller than those of the summer case (case 2; a *mean* of 127 m, a *stddev* of 49 m). Note
this statistical conclusion was also true for each of the four developing stages. Besides, the former case had respective
percentages of 8.5% and 7.5 % of EZT falling into the subranges of 0-50 m and >150 m, while the latter case had respective
percentages of 2.0% and 31 % of EZT falling into the same corresponding subranges. The reason of larger statistical EZT
data (*mean* and *stddev*) and higher percentage (possibility) of larger EZT values (>150 m) for the latter case is attributed to
the stronger solar radiation reaching earth surface in summer than in winter. Stronger solar radiation generally results in
more vigorous and frequent thermals overshooting to higher heights (updrafts) and then moving back (downdrafts).
Consequently entrainments take place in larger vertical regions. Hence both the statistical EZT data (*mean* and *stddev*) and
possibility of larger EZT value seem to provide measures of entrainment intensity. There were also common statistical
characteristics for the two observational cases. For example, the growth stage always had the largest *mean* and *stddev* of
EZT; as neither the NBL nor the FA restricts the booming development of the CBL in the growth stage, the entrainments
were allowed to occur in a wider vertical range. Besides, the quasi-stationary stage usually had the smallest *stddev* of EZT;
this quantitatively reflected the fact that the CBL depth and the EZT changed little in this stage. For all four stages, most
EZT values fell into the 50-150 m subrange; the corresponding overall percentages of EZT falling into the 50-150 m
subrange between 0900 and 1900 LT were 84% and 67% for the winter and summer cases, respectively.
**5 Summary and Conclusions**
A tiled polarization lidar (TPL) has been recently developed for full-day monitoring of the atmospheric boundary layer
(ABL) under various weather conditions. The TPL has a pointing angle of 30 ° off zenith and routinely operates with a time
resolution of 10 s and a range (height) resolution of 7.5 (6.5) m. The equivalent minimum height with full overlap for the





TPL is ~173 m above ground level (AGL). Thus this TPL partly provides a possibility of the depth investigation of shallow
convective boundary layer (CBL) and nocturnal boundary layer (NBL).

From the lidar-recorded range-square corrected elastic signal $X$, the two vertical-distribution-based methods (logarithm
gradient method, LGM; Harr wavelet transform method, HWT) are tested to retrieve instantaneous ABL depth for each $X$
profile. Before applying the LGM and HWT methods, the process-based variance method is first used to locate the hourly-
mean ABL depth. For each given 1-h time interval, the height with maximum variance in the variance profile of aerosol
backscatter ratio (ABR) fluctuations is searched as the hourly-mean ABL depth. By visualizing the time-height contour plots
of $D(z)$ (defined as derivative of logarithmic $X$) and $W_f(z)$ (defined as covariance transform value of $X$), the proper upper
height limits needed for choosing the true height with local maximum variance are intuitive and convenient to be correctly
determined as the base of the misleading aerosol layers aloft. Then the hourly-mean ABL depths provide a guide for an
automated attribution of instantaneous ABL depth by the LGM and HWT methods. A new approach utilizing the full width
at half maximum (FWHM) of the variance profile of ABR fluctuations is developed and proposed to determine the
entrainment zone thickness (EZT). This new approach is designated as FWHM method in this work.

Both a winter and a summer cases of the TPL clear-day measurement results of the CBL and associated entrainment zone
(EZ) are presented. It is concluded that for both typical cases the CBL depth evolution can be described by four consecutive
stages. At the formation stage, the hourly-mean CBL depth grew slowly with a positive growth rate of <0.15 km/h. At the
growth stage, the hourly-mean CBL depth grew fast; the hourly-mean CBL depth growth rate was always >0.3 km/h, and
could reach a value of 0.59 km/h for the summer case. At the quasi-stationary stage, the hourly-mean CBL depth varied
slightly; the hourly-mean CBL depth growth rate changed sign from positive to negative, with absolute value of <0.15 km/h.
At the decay stage, the hourly-mean CBL depth kept decreasing; the hourly-mean ABL depth growth rate could be <-0.4
km/h after sunset. The instantaneous CBL depths exhibited different fluctuation magnitudes in the four stages and the growth
stage always had more obvious fluctuations. The fluctuations of over-plotted ABR profiles in each 1-h time interval also
showed different behaviors at respective stages: the fluctuations usually enlarged at the formation stage, while generally
decreased in the middle part of the CBL at the late growth and quasi-stationary stages. However, the fluctuations of over-
plotted ABR profiles were always prevailing near the top area of the CBL, reflecting the structures of the EZ. The EZT is
subsequently investigated in detail by the proposed FWHM method. It is found that for the same statistical time interval of
0900-1900 LT, the winter case had smaller mean (*mean*) and standard deviation (*stddev*) of EZT data (a *mean* of 94 m, a
*stddev* of 38 m) than those of the summer case (a *mean* of 127 m, a *stddev* of 49 m); besides, the winter case had respective
percentages of 8.5% and 7.5 % of EZT values falling into the subranges of 0-50 m and >150 m, while the summer case had
respective percentages of 2.0% and 31 % of EZT falling into the same corresponding subranges. Common statistical
characteristics also existed for both cases. The growth stage always had the largest *mean* and *stddev* of EZT and the quasi-
stationary stage usually had the smallest *stddev* of EZT. For all four stages, most EZT values fell into the 50-150 m subrange.



The corresponding overall percentages of EZT falling into the 50-150 m subrange between 0900 and 1900 LT are 84% and
67% for the winter and summer cases, respectively.

The proposed FWHM method utilizes the FWHM of the variance profile of the ABR fluctuations to quantify the EZT.
Considering the observed ratios of EZT to CBL depth mostly have values of <20%, the retrieved EZT values seem
reasonable. In future, this method can be verified by comparisons with other approaches (e.g., comparisons with results by
intensive radiosonde). It is also checked that similar characteristics of the four-stage evolution of the CBL and the common
statistics of the associated EZ seem to hold true for other 4 clear-day observations. However, it can be much more
complicated when heavy aerosol loads and clouds are present. Further investigations on the CBL and EZ under various
weather conditions shall be presented in our following works.
**Author contributions**
FL built the lidar system, performed the data analysis and wrote the initial manuscript. FY conceived the project and led the
study. ZY, YZ, YH and YY performed the lidar observations, glued the raw data and participated in scientific discussions.
All authors discussed the results and finalized the manuscript.
**Competing interests**
The authors declare that they have no conflict of interest.
**Data availability**
Lidar data used in this work are available under permission (yf@whu.edu.cn).
**Acknowledgments**
This work was funded by the National Natural Science Foundation of China under Grant 41927804. The authors would like
to express thanks to Yifan Zhan for discussions on ABL depth retrieving algorithms and to Xiangliang Pan and Wei Wang
for lidar calibration experiments and data collections.

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



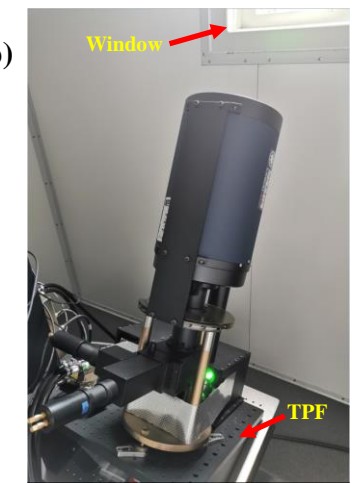

**Figure 1: (a) Schematic optical layout of the TPL. PR, polarizer; BE, beam expander; RM, reflecting mirror; CL, collimating lens; HWP, half-wavelength plate; IF, interference filter; PBS, polarization beam splitter; FL, focusing lens; PMT, photomultiplier tube; (b) a picture of the lidar optics. The whole optics is placed on a tilted platform (TPF). A window permits propagating laser beam and atmospheric backscatter to pass through without blocking.**



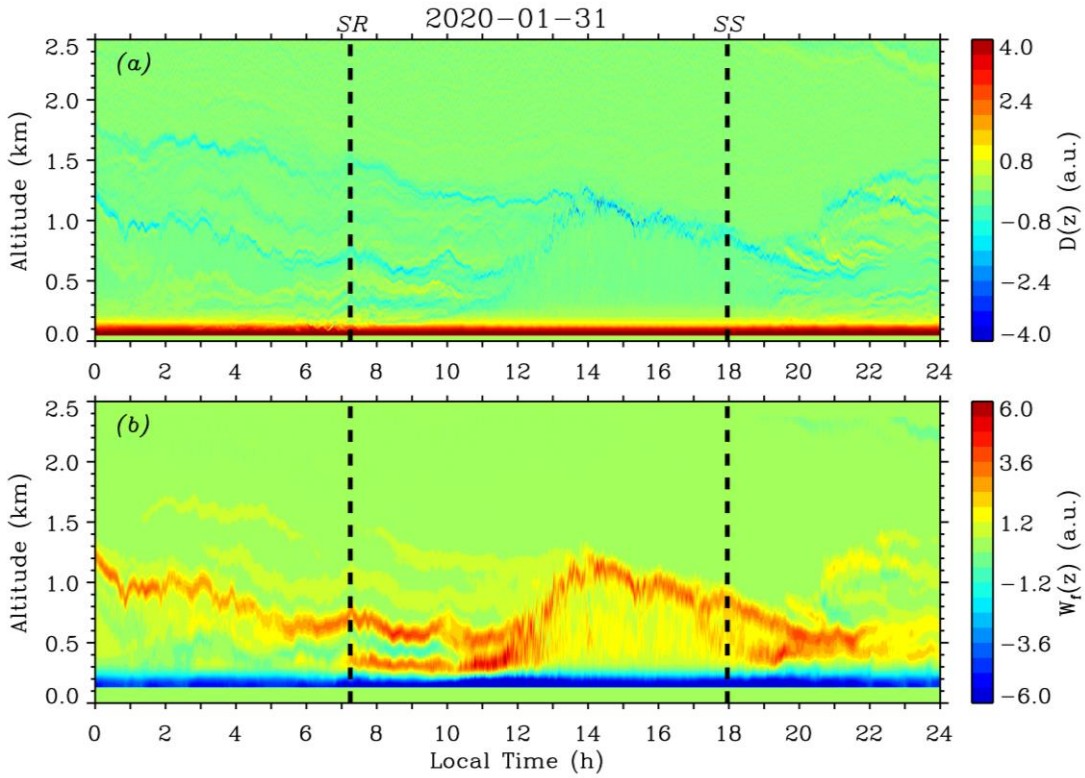



**Figure 2: Contour plots of (a) $D(z)$ and (b) $W_f(z)$ on Jan 31, 2020. Sunrise (SR) and sunset (SS) times are marked by**
**thick black dashed lines. Multiple (residual) aerosol layers which definitely lead to misattribution of ABL depth, are**
**clearly indicated by stripes of local minima of $D(z)$ and maxima of $W_f(z)$ in the contour plots. By visualizing these**
**contour plots, proper upper heights for applying the variance method can be conveniently and correctly determined**
**to be below the base of multiple (residual) aerosol layers aloft.**










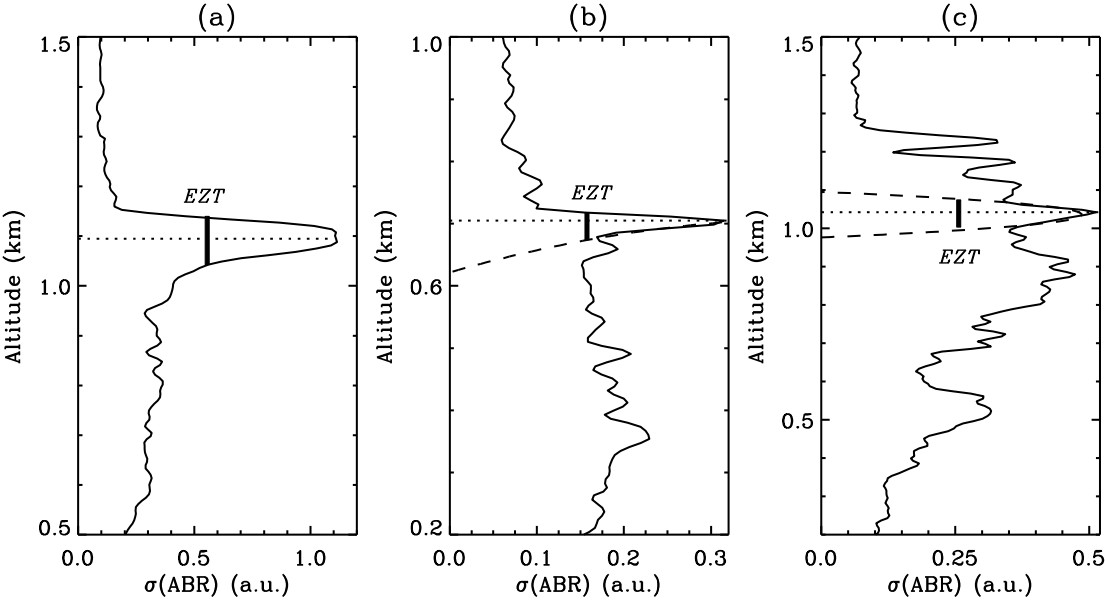



**Figure 3: Illustrations of the FWHM method using the variance of ABR fluctuations to determine the CBL depth and subsequent EZT. Thin black lines indicate the standard deviation of ABR fluctuations, $\sigma$(ABR). Thin dotted lines specify the CBL depth with maximum $\sigma$(ABR). Thick vertical lines represent the determined EZT (EZ). (a) For a strong updraft case, both the upper and lower edges near the peak $\sigma$(ABR) are clear-cut and steep. The EZT can be directly obtained; (b) for a less-intense updraft case, the lower edge is not clear-cut enough. A quadratic polynomial fitting (dashed line) is applied to the lower edge to help determine the EZT; (c) for a weak turbulence and advected aerosol case, neither the upper nor the lower edge is clear-cut enough. Quadratic polynomial fittings (dashed lines) are applied to both edges to help determine the EZT.**












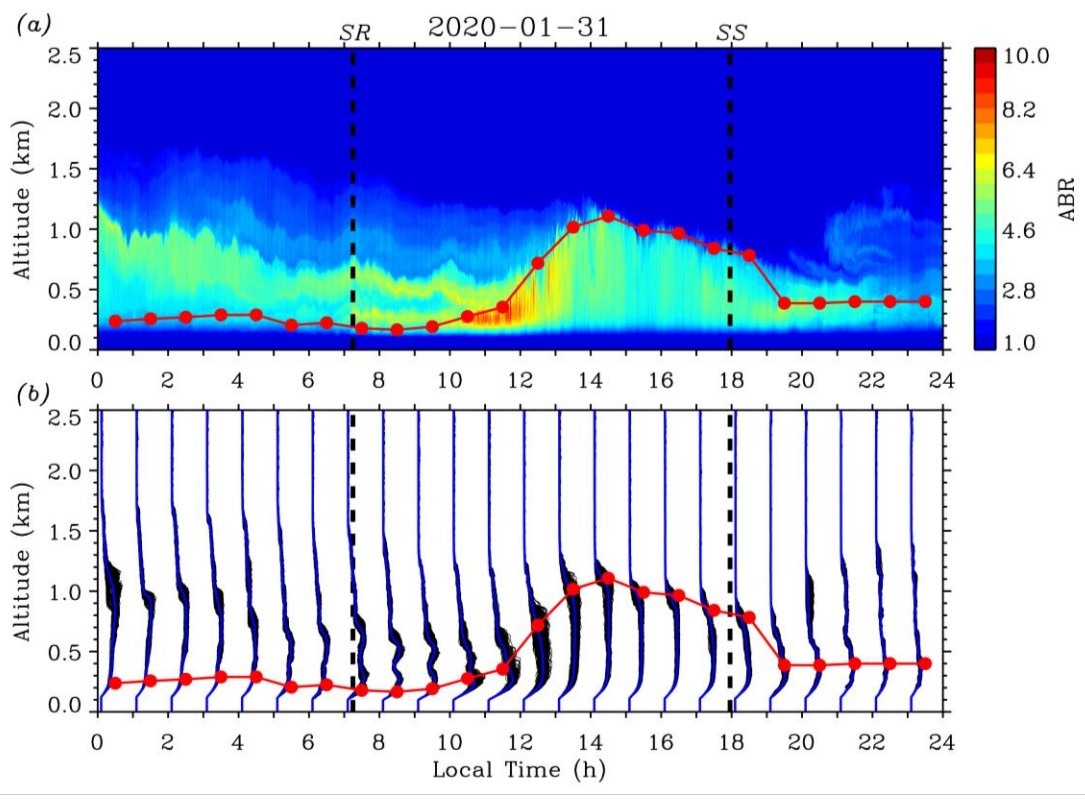



**Figure 4: (a) Contour plot of the ABR on Jan 31, 2020; (b) over-plots of ABR profiles (thin black lines) in each 1-h**
**time interval and the hourly mean ABR profile (blue line). SR and SS times are indicated by thick black dashed lines.**
**Red solid circle represents the hourly mean ABL depth retrieved by the variance method and the red line indicates**
**the diurnal evolution trend of the ABL depth.**









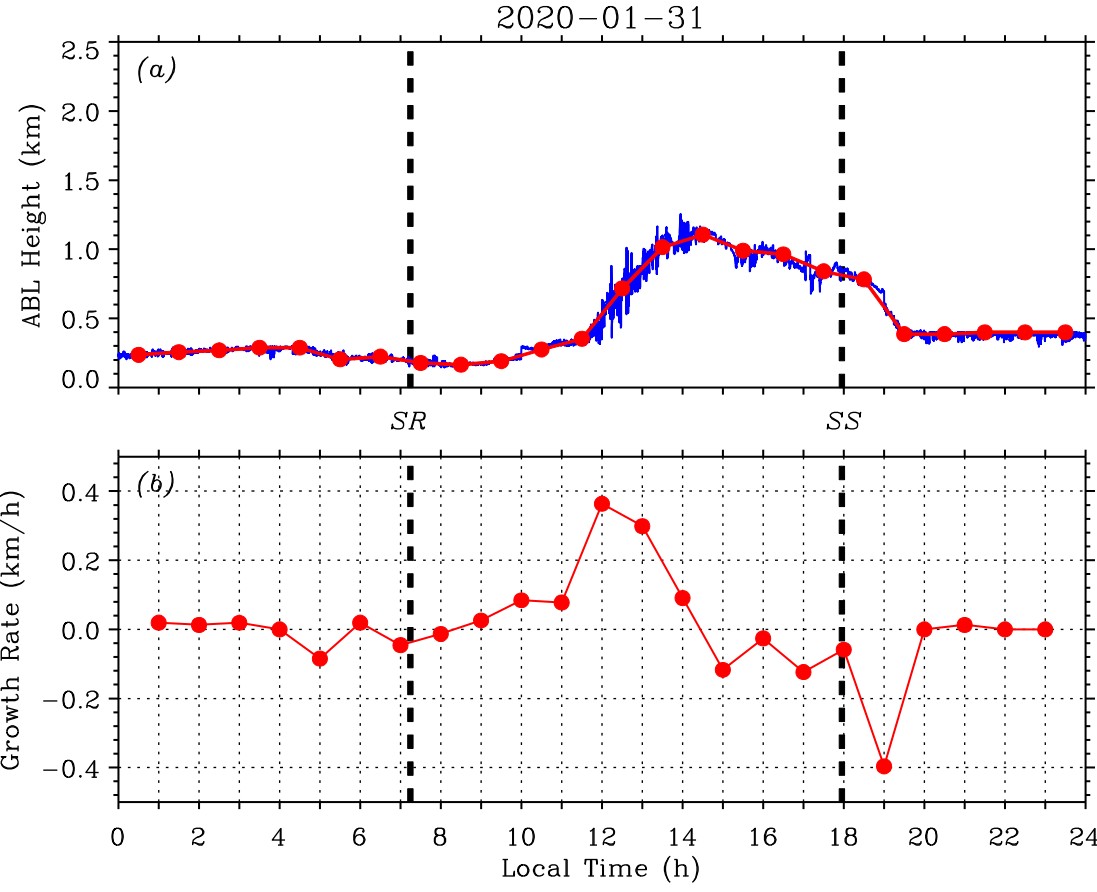

**Figure 5: (a) instantaneous ABL depths (blue) obtained by LGM method (before 1000 and after 1900 LT) and HWT method (between 1000 and 1900 LT). Red solid circles indicate the hourly mean ABL depth by variance method; (b) hourly mean ABL depth growth rate. Thick black dashed lines mark the SR and SS times on Jan 31, 2020.**



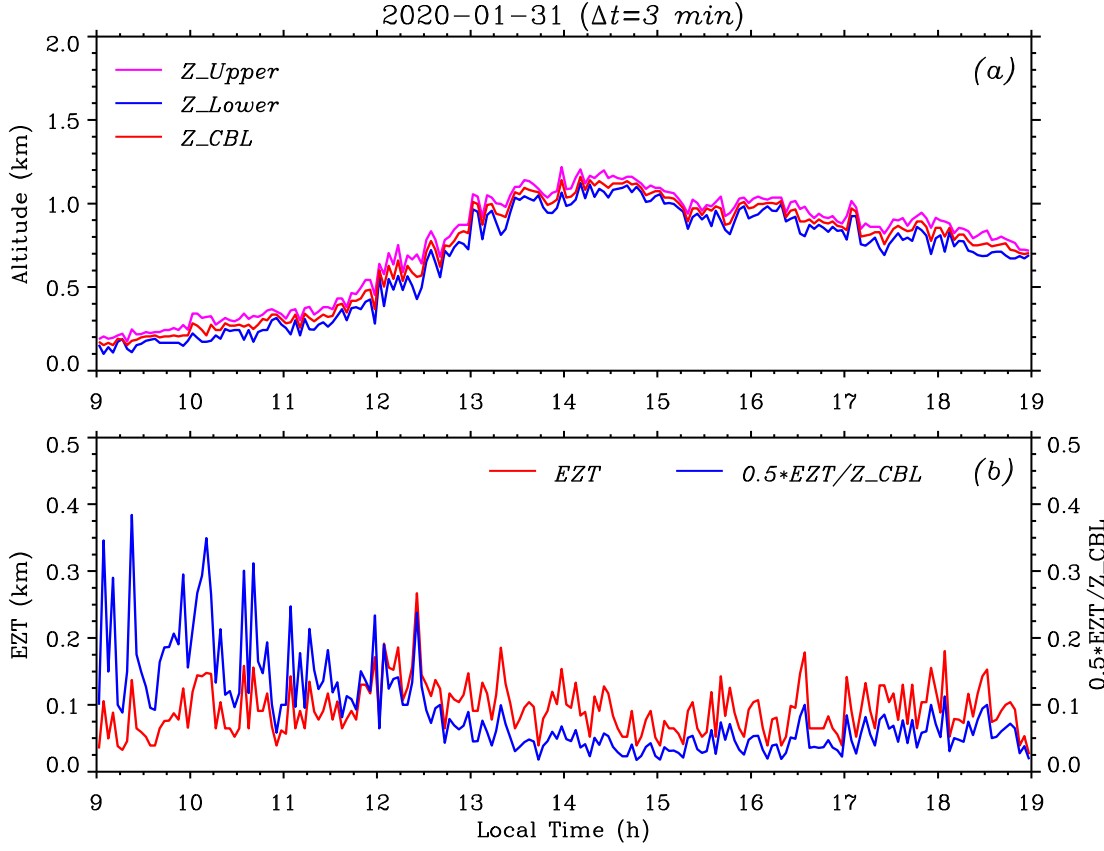



**Figure 6: (a) The CBL depth $Z\_CBL$ (red) obtained by the variance method between 0900 and 1900 LT on Jan 31,**
**2020. The EZ upper height $Z\_Upper$ (magenta) and lower height $Z\_Lower$ (blue) are derived from the FWHM of the**
**σ(ABR) profile each of which is calculated within a time interval of 3 min; (b) corresponding EZT (red) and ratio of**
**EZT to $Z\_CBL$ (blue) during the same time interval. Note the ratio is multiplied by a factor of 0.5 so that the two**
**vertical axes share the same scaling range.**












**Table 1: Statistics of EZT obtained on Jan 31, 2020**

| Stage of CBL | | Formation | Growth | Quasi-stationary | Decay | Total |
|---|---|---|---|---|---|---|
| Time Interval (LT) | | 0900-1130 | 1130-1330 | 1330-1630 | 1630-1900 | 0900-1900 |
| Statistical data of EZT(km) | *min* | 0.033 | 0.065 | 0.039 | 0.026 | 0.026 |
| | *max* | 0.158 | 0.267 | 0.154 | 0.180 | 0.267 |
| | *mean* | 0.085 | 0.122 | 0.082 | 0.095 | 0.094 |
| | *stddev* | 0.036 | 0.041 | 0.028 | 0.036 | 0.038 |
| Percentages in each EZT subrange (%) | 0.00-0.05 km | 16.0 | 0.0 | 10.0 | 6.0 | 8.5 |
| | 0.05-0.10 km | 54.0 | 27.5 | 65.0 | 52.0 | 51.5 |
| | 0.10-0.15 km | 26.0 | 52.5 | 23.3 | 34.0 | 32.5 |
| | 0.15-0.20 km | 4.0 | 17.5 | 1.7 | 8.0 | 7.0 |
| | 0.20-0.30 km | 0.0 | 2.5 | 0.0 | 0.0 | 0.5 |



















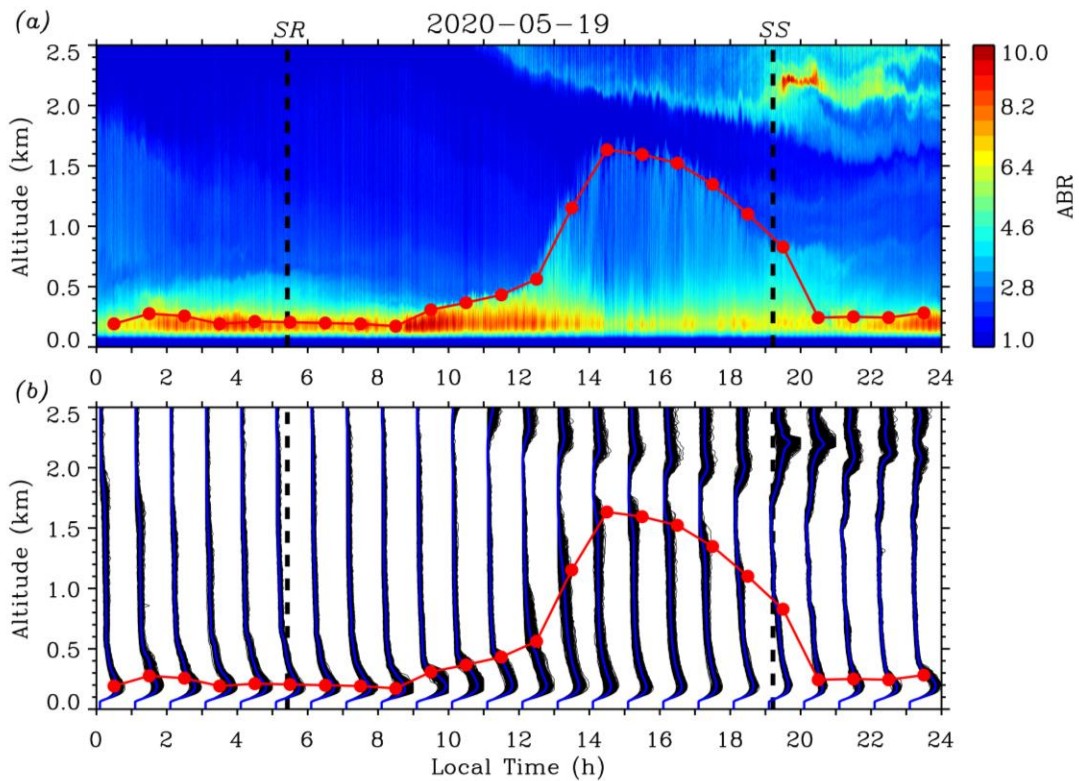



**Figure 7: Same as Figure 4 but on the day of May 19, 2020.**


















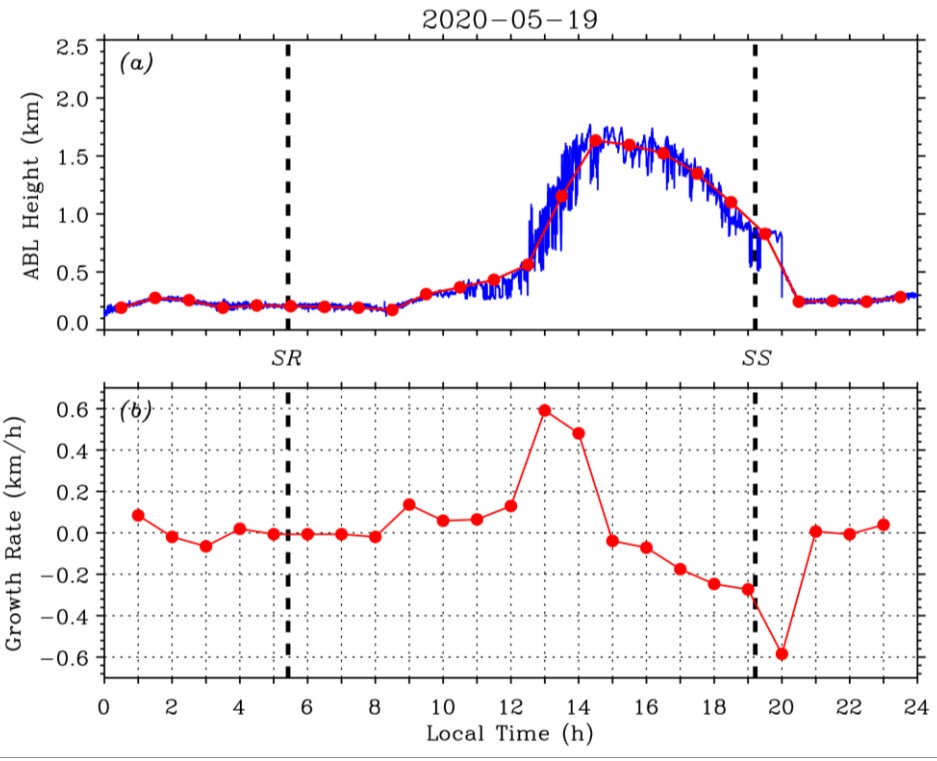



**Figure 8: Same as Figure 5 but on the day of May 19, 2020.**













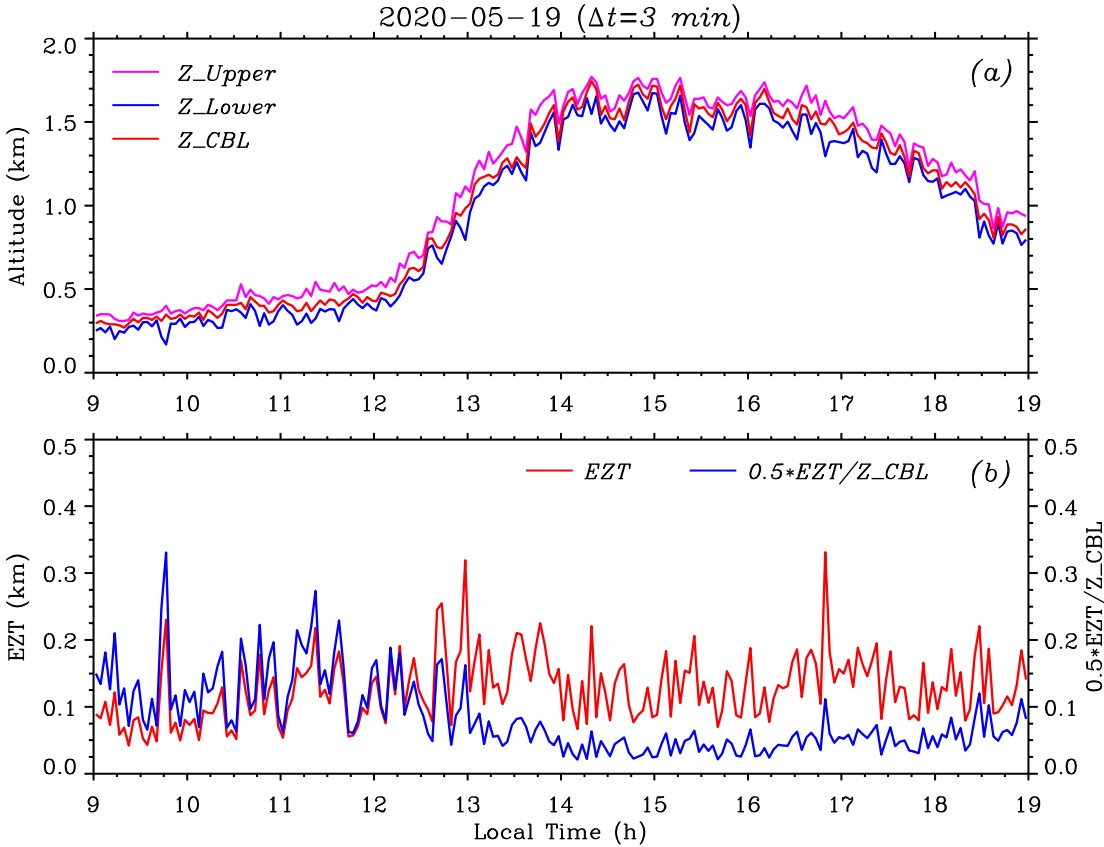

**Figure 9: Same as Figure 6 but on the day of May 19, 2020.**








**Table 2: Statistics of EZT obtained on May 19, 2020**

| Stage of CBL | | Formation | Growth | Quasi-stationary | Decay | Total |
|---|---|---|---|---|---|---|
| Time Span (LT) | | 0900-1230 | 1230-1430 | 1430-1630 | 1630-1900 | 0900-1900 |
| Statistical data of EZT(km) | *min* | 0.042 | 0.066 | 0.070 | 0.079 | 0.042 |
| | *max* | 0.230 | 0.319 | 0.206 | 0.331 | 0.331 |
| | *mean* | 0.106 | 0.153 | 0.122 | 0.142 | 0.127 |
| | *stddev* | 0.044 | 0.057 | 0.035 | 0.046 | 0.049 |
| Percentages in each EZT subrange (%) | 0.00-0.05 km | 5.7 | 0 | 0 | 0 | 2.0 |
| | 0.05-0.10 km | 50.0 | 20.0 | 35.0 | 20.0 | 33.5 |
| | 0.10-0.15 km | 25.7 | 32.5 | 40.0 | 40.0 | 33.5 |
| | 0.15-0.20 km | 15.7 | 27.5 | 22.5 | 36.0 | 24.5 |
| | 0.20-0.34 km | 2.9 | 20.0 | 2.5 | 4.0 | 6.5 |





















**Table 3: Comparisons of EZT statistics for the winter and summer cases**

| Case 1 (Jan 31, 2020) | | Formation | Growth | Quasi-stationary | Decay | Total |
|---|---|---|---|---|---|---|
| Time Span (LT) | | 0900-1130 | 1130-1330 | 1330-1630 | 1630-1900 | 0900-1900 |
| Statistical data (km) | *mean* | 0.085 | 0.122 | 0.082 | 0.095 | 0.094 |
| | *stddev* | 0.036 | 0.041 | 0.028 | 0.036 | 0.038 |
| Percentages (%) | 0.00-0.05 km | 16.0 | 0.0 | 10.0 | 6.0 | 8.5 |
| | 0.05-0.15 km | 80.0 | 80.0 | 88.3 | 86.0 | 84.0 |
| | 0.15-0.30 km | 4.0 | 20.0 | 1.7 | 8.0 | 7.5 |
| Case 2 (May 19, 2020) | | Formation | Growth | Quasi-stationary | Decay | Total |
| Time Span (LT) | | 0900-1230 | 1230-1430 | 1430-1630 | 1630-1900 | 0900-1900 |
| Statistical data (km) | *mean* | 0.106 | 0.153 | 0.122 | 0.142 | 0.127 |
| | *stddev* | 0.044 | 0.057 | 0.035 | 0.046 | 0.049 |
| Percentages (%) | 0.00-0.05 km | 5.7 | 0 | 0 | 0 | 2.0 |
| | 0.05-0.15 km | 75.7 | 52.5 | 75.0 | 60.0 | 67.0 |
| | 0.15-0.34 km | 18.6 | 47.5 | 25.0 | 40.0 | 31.0 |

