# Peer review of "Measurement report: Characteristics of clear-day convective"

_Atmospheric Chemistry and Physics, 2020_

## Referee Comment (RC1) · Anonymous Referee #3 · 20 Nov 2020

This manuscript presented case studies of convective boundary layer (CBL) and entrainment zone observed by a ground-based lidar. The evolution of CBL has been described by four stages. The values of CBL depth and entrainment zone thickness (EZT) are reported under different stages. However, the paper only discusses a few cases. Meanwhile, the meaning and significance of this study are not clear. Therefore, this paper needs major revisions before publication.

Specific Comments:

1. The characteristics of CBL and entrainment zone are widely reported in numerous previous papers. I do not find the new characteristics of CBL in this study. The authors

may carefully consider the title. The title also should include the measurement location (Wuhan).

2. The introduction part needs improvements. Currently, this section introduced some related works, but did not state the limitations in previous studies. This section also did not tell readers the novelty of this work.

3. The manuscript classified the evolution of CBL into four stages. However, it is a well-known feature, which is well discussed by Stull. (1988). I suggest the authors refer such classifications to the previous papers.

4. The determination of EZT is a highlight in this study. Nonetheless, there is a lack of validations of EZT retrievals derived from FWHM. The limited cases also cannot support the robustness of this method.

5. Page 15, line 457. The statement is not appropriate. The ratio of EZT to CBL depth cannot support the accuracy of the retrieved EZT values.

6. The authors may consider revising the manuscript type as "Measurement Report", which more fit the scope of this study.

References.

Stull, R. B.: An Introduction to Boundary Layer Meteorology, Kluwer Acad. Publ., Netherlands, 1988.
* * *

---

## Referee Comment (RC2) · Anonymous Referee #1 · 29 Nov 2020

Review of Characteristics of convective boundary layer and associated entrainment zone as observed by a ground-based polarization lidar by Liu et al.

General Comments:

Entrainment is critical for the evolution of boundary layer. This study developed an approach for estimating entrainment zone thickness. Then this approach was applied to two cases. The evolution of boundary layer and entrainment zone thickness was analyzed at four stages. The difference between the winter and summer cases were also discussed. The topic is interesting but major revision is needed before I can recommend acceptance of this paper.

1. Line 216-217: "Then, the upper and lower heights with half value of the maximum variance are searched and defined as the top and bottom heights of EZ, respectively." Why do the upper and lower heights with half value of the maximum variance represent the top and bottom heights of EZ? Please compare the top and bottom of EZ from this method with those from other methods to justify this method.

2. Only two cases are analyzed to represent the results in winter and summer, respectively. To obtain robust conclusions, more cases are needed, at least, one month for each season. In addition, why do the authors only focus on winter and summer? Please include spring and autumn. The case on May 19, 2020 is actually a case in spring, not summer.

3. In section 4, only the results from this study are presented. Please compare these results with previous studies.

Minor Comments:

1. In the introduction, please clearly state what is the deficiency of previous studies on this topic and what is new in this study.

2. Line 214: ABR should be defined when it shows up for the first time.

3. Line 242-246: Please give a figure to compare the ABR results by this TPL and the co-located vertically-pointing 532-nm polarization lidar.

---

## Referee Comment (RC3) · Anonymous Referee #2 · 6 Dec 2020

This manuscript is one part of an increasingly long list of papers simultaneously investigating the evolution of both planetary boundary layer (PBL) height and entrainment zone thickness for the haze events in China. Most have focused on CBL, this is one of the few to deal with EZ. It is of essence to investigate the variation of EZT near the PBL top, since it concerns the formation of cloud, the interaction of land-atmosphere, and the vertical mixing of scalars. The retrieval methods are scientifically robust, and the results interpretation makes sense, as far as I can tell. Therefore, I recommend acceptance for publication after addressing the following concerns.

Major comments: 1. The title of this manuscript seems overstated. Actually, the au-

thors only dealt with two cases from lidar measurements in Wuhan. Therefore, the title should be revised.

2. Most of the sentences are almost the same in both Conclusion and Abstract, especially regarding the statistic results of EZT evolutioin at different stages for both winter and summer cases. This should be avoided. The authors are suggested to highlight the major findings as well as the importance or implications of their work in Abstract, rather than simply duplicating the numbers. Minor comments: L17: the first "FWHM" is redudant and can be deleted. L57: "despite of" -> "despite" L77: "EZT" is a geophysical parameter rather than an approach. The authors mean "the determination of EZT"? L181: "Jan" is not official acronym for "January", and should be given full spelling. All instrances should be corrected throughout the MS L193: "measuring"-> "measurement" L245:"convinced"-> "confirmed" L432: "This new approach is designated as FWHM method in this work." can be deleted.

---

## Author Comment (AC1) · 3 Jan 2021

Referee's Comments: This manuscript presented case studies of convective boundary layer (CBL) and entrainment zone observed by a ground-based lidar. The evolution of CBL has been described by four stages. The values of CBL depth and entrainment zone thickness (EZT) are reported under different stages. However, the paper only discusses a few cases. Meanwhile, the meaning and significance of this study are not clear. Therefore, this paper needs major revisions before publication.

Authors' response: We greatly appreciate this Referee for the thoughtful considerations and pertinent comments on the current manuscript. Following the Referee's constructive suggestions, we have added another two typical clear-day cases; besides, we have revised the Abstract and Introduction parts to point out the meaning and significance of this study. In response to the Referee's concerns, all necessary modifications are made point by point in the revised manuscript.

Specific Comments: 1. The characteristics of CBL and entrainment zone are widely reported in numerous previous papers. I do not find the new characteristics of CBL in this study. The authors may carefully consider the title. The title also should include the measurement location (Wuhan).

Authors' response: We thank the Referee for the suggestion of a more appropriate title for concluding the current work. Now the title has been changed to "Characteristics of clear-day convective boundary layer and associated entrainment zone as observed by a ground-based polarization lidar over Wuhan ( $30.5^{\circ}N$ ,  $114.4^{\circ}E$ )" to state that the conclusions are limited to clear-day weather conditions and to the observational location of Wuhan ( $30.5^{\circ}N$ ,  $114.4^{\circ}E$ ).

The CBL depth is analyzed in this study according to its four evolution stages and it is found "The instantaneous CBL depths exhibited different fluctuation magnitudes in the four stages and fluctuations at the growth stage were generally larger", which we believe is distinctive from previous studies.

2. The introduction part needs improvements. Currently, this section introduced some related works, but did not state the limitations in previous studies. This section also did not tell readers the novelty of this work.

Authors' response: We thank the Referee for the constructive comments on the introduction part. Along these valuable suggestions, the sentences "However, the above two introduced methods yield EZT values with large differences (e.g., Pal et al., 2010); the choice of specific percentages of air having the FA characteristics for the definition of EZ bottom height is variable (between 5% and 15%) among different researchers (e.g., Deardorff et al., 1980; Wilde et al., 1985; Flamant et al., 1997; Cohn **ACPD**
and Angevine, 2000; Pal et al., 2010). Moreover, considering that variations of ABL depths can result from not only entrainment but also non-turbulent processes (e.g., atmospheric gravity waves and mesoscale variations in ABL structure), the methods depending on variations of ABL depth might not really characterize the true EZ (Davis et al., 1997). So far, no universally accepted approach exists for the determination of EZT (Brooks and Fowler, 2007)" are added to review on the limitations of the current EZT determination approaches. Besides, the last paragraph of the introduction part now reads "Currently, studies are generally concentrated on the CBL while relatively rare on the EZ. The basic physical processes governing entrainment and their relationship with other boundary layer properties are still not fully understood (Brooks and Fowler, 2007). Besides, the general grid increments of state-of-the-art weather forecast and climate models are too coarse to resolve small-scale boundary layer turbulence (Wulfmeyer et al., 2016). Therefore, continuous and high-resolution measurements at various observational locations to infer detailed knowledge on both CBL and associated EZ, especially small-scale boundary layer turbulence therein, are of significant importance to boundary layer related studies including land-atmosphere interaction, air quality forecast and almost all weather and climate models (Wulfmeyer et al., 2016). In this work we present the high-resolution measurement results of the CBL and associated EZ using a recently-developed titled polarization lidar (TPL) over Wuhan (30.5°N, 114.4°E). The TPL is housed in a specially-customized working container and capable of operating under various weather conditions (including heavy precipitation). The TPL has an inclined working angle of 30° off zenith and routinely monitors the atmosphere with a time resolution of 10 s and a height resolution of 6.5 m. The equivalent minimum height with full overlap for the TPL is  $\sim$ 173 m above ground level (AGL). Based on the TPL-measured backscatter, a new approach has been developed for determination of the EZT. The small-scale characteristics of the CBL and associated EZ have also been investigated which can contribute to the improvement of understanding the structures and variations of the ABL, as well as parameterization of the EZ. The instrument, methodology, observational results and summary and conclusions are stated succes-

**ACPD**
sively in following sections" to state the meaning, significance and novelty of this work. We feel that the introduction part has been greatly improved after modification.

3. The manuscript classified the evolution of CBL into four stages. However, it is a well-known feature, which is well discussed by Stull. (1988). I suggest the authors refer such classifications to the previous papers.

Authors' response: Following the Referee's suggestion, the excellent work by Stull (1988) has now been referred to in the revised manuscript for classifying the evolution of CBL into four stages.

4. The determination of EZT is a highlight in this study. Nonetheless, there is a lack of validations of EZT retrievals derived from FWHM. The limited cases also cannot support the robustness of this method.

Authors' response: We thank the Referee for suggesting validation of the EZT from the FWHM method. We believe this FWHM method to be physically sound as it directly reflects the mixing history of the aerosols (tracer) in the EZ. However, direct validation of the EZT retrievals is difficult as reviewed in the Introduction "So far, no universally accepted approach exists for the determination of EZT" and the existing approaches have their own deficiencies. A comparison with EZT result determined by its theoretical definition that corresponds to the vertical region with mean negative buoyancy flux might be favoured in future. Besides, two more clear-day cases are added in the revised manuscript to support the robustness of this method.

Along the Referee's suggestions, a special paragraph is now added to discuss on this issue in an added subsection "4.3 Discussion on the clear-day EZT statistics and the FWHM method". It reads "Note the proposed FWHM method utilizes the FWHM of the variance profile of the ABR fluctuations to quantify the EZT. We believe it to be physically sound as it directly reflects the mixing history of aerosols (tracer) in the EZ. When applying it to lidar data, it definitely determines the EZ (and consequently the EZT) when turbulence is dominating and the variance profile of ABR fluctuations has clear-

ACPD
cut edges. However, caution must be taken when turbulence is weak and the variance profile of ABR fluctuations suffers from interference of residual layer and/or advected aerosols. The retrieved EZT values for the four typical clear-day cases mostly fall into the 50-150 m range with a percentage of  $\geq$ 67%, while the overall EZT values range from 0 to 340 m. Pal et al. (2010) reported the lidar-derived EZT retrievals for a summer case using the cumulative frequency distribution method, which had mean values of 75 m and 62 m and magnitude ranges of 10-230 m and 0-200 m for the guasi-stationary and growth stages, respectively. While for the early autumn case in this work, the EZT results had mean values of 113 m and 123 m and magnitude ranges of 41-279 m and 39-289 m for the guasi-stationary and growth stages, respectively. These observational results differ obviously for the mean EZT values and magnitude ranges. But this comparison seems not rigorous as the EZT results were obtained at distinct observational locations. For a better validation of the reliability of the FWHM approach, comparisons with EZT values retrieved by co-located intensive radiosonde or by synergy of highresolution temperature lidar (Behrendt et al., 2015) and Doppler lidar (Ansmann et al., 2010), in which situation the EZT might be determined by its theoretical definition that corresponds to the vertical region with mean negative buoyancy flux (Driedonks and Tenneke, 1984; Cohn and Angevine, 2000), shall be favoured in the future".

5. Page 15, line 457. The statement is not appropriate. The ratio of EZT to CBL depth cannot support the accuracy of the retrieved EZT values.

Authors' response: We agree with the Referee and now the sentence "Considering the observed ratios of EZT to CBL depth mostly have values of

Please also note the supplement to this comment: https://acp.copernicus.org/preprints/acp-2020-963/acp-2020-963-AC1-supplement.pdf

---

## Author Comment (AC2) · 3 Jan 2021

General Comments: Entrainment is critical for the evolution of boundary layer. This study developed an approach for estimating entrainment zone thickness. Then this approach was applied to two cases. The evolution of boundary layer and entrainment zone thickness was analyzed at four stages. The difference between the winter and summer cases were also discussed. The topic is interesting but major revision is needed before I can recommend acceptance of this paper.

Authors' response: We greatly thank this Referee for the thorough reading of the manuscript and encouraging comments on the current work. According to the Referee's valuable suggestions, all necessary modifications are made point by point in the revised manuscript.

Major comments: 1. Line 216-217: "Then, the upper and lower heights with half value of the maximum variance are searched and defined as the top and bottom heights of EZ, respectively." Why do the upper and lower heights with half value of the maximum variance represent the top and bottom heights of EZ? Please compare the top and bottom of EZ from this method with those from other methods to justify this method.

3. In section 4, only the results from this study are presented. Please compare these results with previous studies.

Authors' response: (a) Along the Referee's suggestion, a new sentence "Note here the FWHM of the variance profile of ABR fluctuations is utilized because it physically represents that most aerosols have been strongly mixed in the vertical height interval defined according to the FWHM" has been added subsequently to state why the upper and lower heights with half value of the maximum variance represent the top and bottom heights of EZ (Please see Line 238 in the revised manuscript).

(b) We thank the Referee for suggesting validation of the EZT from the FWHM method. We believe this FWHM method to be physically sound as it directly reflects the mixing history of the aerosols (tracer) in the EZ. However, direct validation of the EZT retrievals is difficult as reviewed in the Introduction "So far, no universally accepted approach exists for the determination of EZT" and the existing approaches have their own deficiencies. A comparison with EZT result determined by its theoretical definition that corresponds to the vertical region with mean negative buoyancy flux might be favoured in future. In response to the Referee's suggestions, a special paragraph including comparison with previous studies is now added to discuss on this issue in an added subsection "4.3 Discussion on the clear-day EZT statistics and the FWHM method". It reads "Note the proposed FWHM method utilizes the FWHM of the variance profile of the ABR fluctuations to quantify the EZT. We believe it to be physically sound as it

directly reflects the mixing history of aerosols (tracer) in the EZ. When applying it to lidar data, it definitely determines the EZ (and consequently the EZT) when turbulence is dominating and the variance profile of ABR fluctuations has clear-cut edges. However, caution must be taken when turbulence is weak and the variance profile of ABR fluctuations suffers from interference of residual layer and/or advected aerosols. The retrieved EZT values for the four typical clear-day cases mostly fall into the 50-150 m range with a percentage of ≥67%, while the overall EZT values range from 0 to 340 m. Pal et al. (2010) reported the lidar-derived EZT retrievals for a summer case using the cumulative frequency distribution method, which had mean values of 75 m and 62 m and magnitude ranges of 10-230 m and 0-200 m for the quasi-stationary and growth stages, respectively. While for the early autumn case in this work, the EZT results had mean values of 113 m and 123 m and magnitude ranges of 41-279 m and 39-289 m for the quasi-stationary and growth stages, respectively. These observational results differ obviously for the mean EZT values and magnitude ranges. But this comparison seems not rigorous as the EZT results were obtained at distinct observational locations. For a better validation of the reliability of the FWHM approach, comparisons with EZT values retrieved by co-located intensive radiosonde or by synergy of high-resolution temperature lidar (Behrendt et al., 2015) and Doppler lidar (Ansmann et al., 2010), in which situation the EZT might be determined by its theoretical definition that corresponds to the vertical region with mean negative buoyancy flux (Driedonks and Tenneke, 1984; Cohn and Angevine, 2000), shall be favoured in the future".

2. Only two cases are analyzed to represent the results in winter and summer, respectively. To obtain robust conclusions, more cases are needed, at least, one month for each season. In addition, why do the authors only focus on winter and summer? Please include spring and autumn. The case on May 19, 2020 is actually a case in spring, not summer.

Authors' response: We thank the Referee for suggesting more cases to yield robust conclusions. Two more clear-day cases have now been added in the revised

manuscript. Besides, the case on May 19, 2020 is renamed as a later spring case. It is a pity that we failed to find a suitable clear-day case in summer months (June, July and August) due to rainy, and/or patchy-cloudy weather conditions. Instead, an early autumn case (on September 7, 2020) is selected as representative of a summer case since the surface temperatures (21-34 °C) on this day were comparable with those on summer days (20-37 °C; please refer to Table S3 in the Supplement for detail).

In response to the Referee's constructive suggestion, a new discussion subsection (4.3 Discussion on the clear-day EZT statistics and the FWHM method) has been added in the revised manuscript. Now the corresponding sentences read "In combination with the above-two presented typical cases, another two clear-day cases (on the days of September 7 and November 12, 2020, respectively) are also investigated to demonstrate the robustness of the FWHM method and the representativeness of the conclusions on the EZ. The corresponding contour plots of the ABR, plots of the ABL depth and EZT evolution, as well as tables of obtained EZT statistics, are provided in the Supplement. Since no suitable clear-day case is available for the summer days of 2020 due to rainy and/or patchy-cloudy weather conditions, the early autumn result on September 7, 2020 is selected here and regarded as representative of a summer case as the surface temperatures on this day (21-34 °C) were comparable with those on summer days (20-37 °C; see Table S3 in the Supplement). Table 3 compares the EZT statistics for all the four picked cases.

As shown in Table 3, all four cases exhibited apparent statistical differences. For the same time interval of 0900-1900 LT, the winter case (case 1; a mean of 94 m, a stddev of 38 m) and the late autumn case (case 4; a mean of 103 m, a stddev of 48 m) had overall statistical EZT data smaller than those of the late spring case (case 2; a mean of 127 m, a stddev of 49 m) and the early autumn case (case 3; a mean of 113 m, a stddev of 60 m). Note this statistical conclusion was also true for each of the four developing stages. Besides, the winter case (8.5%) and the late autumn case (11.5%) had larger percentages of EZT falling into the subranges of 0-50 m than those of the late

spring case (2.0%) and the summer case (8.0%), but smaller percentages (7.5% and 18.0%, respectively) of EZT falling into the subranges of >150 m compared to those of the late spring case (31.0%) and the summer case (24.0%). The reason of larger EZT statistics (mean and stddev) and higher percentage (possibility) of larger EZT values (>150 m) for the late spring and early autumn cases is attributed to the stronger solar radiation reaching the earth surface in late spring/early autumn than in winter/late autumn (Guo et al., 2020). Stronger solar radiation generally results in more vigorous and frequent thermals overshooting to higher heights (updrafts) and then moving back (downdrafts). Consequently entrainments take place in larger vertical regions. Hence both the EZT statistics (mean and stddev) and possibility of larger EZT value seem to provide measures of entrainment intensity. There were also common characteristics for the four observational cases. For example, all four cases showed moderate variations of mean of EZT from stage to stage. The growth stage always had the largest mean and stddev of EZT; as neither the NBL nor the FA restricts the booming development of the CBL in the growth stage, the entrainments were allowed to occur in a wider vertical range. Besides, the quasi-stationary stage usually had the smallest stddev of EZT; this quantitatively reflected the fact that the CBL depth and the EZT changed little in this stage. For all four stages, most EZT values fell into the 50-150 m subrange; the corresponding overall percentages of EZT falling into the 50-150 m subrange between 0900 and 1900 LT were 84%, 67%, 68% and 70.5% for the winter, late spring, early autumn and late autumn cases, respectively."

Minor Comments: 1. In the introduction, please clearly state what is the deficiency of previous studies on this topic and what is new in this study.

Authors' response: We really appreciate the Referee for the constructive suggestions on the Introduction part. Along these valuable suggestions, the sentences "However, the above two introduced methods yield EZT values with large differences (e.g., Pal et al., 2010); the choice of specific percentages of air having the FA characteristics for the definition of EZ bottom height is variable (between 5% and 15%) among different

researchers (e.g., Deardorff et al., 1980; Wilde et al., 1985; Flamant et al., 1997; Cohn and Angevine, 2000; Pal et al., 2010). Moreover, considering that variations of ABL depths can result from not only entrainment but also non-turbulent processes (e.g., atmospheric gravity waves and mesoscale variations in ABL structure), the methods depending on variations of ABL depth might not really characterize the true EZ (Davis et al., 1997). So far, no universally accepted approach exists for the determination of EZT (Brooks and Fowler, 2007)" are added to review on the limitations of the current EZT determination approaches. Besides, the last paragraph of the Introduction part now reads "Currently, studies are generally concentrated on the CBL while relatively rare on the EZ. The basic physical processes governing entrainment and their relationship with other boundary layer properties are still not fully understood (Brooks and Fowler, 2007). Besides, the general grid increments of state-of-the-art weather forecast and climate models are too coarse to resolve small-scale boundary layer turbulence (Wulfmeyer et al., 2016). Therefore, continuous and high-resolution measurements at various observational locations to infer detailed knowledge on both CBL and associated EZ, especially small-scale boundary layer turbulence therein, are of significant importance to boundary layer related studies including land-atmosphere interaction, air quality forecast and almost all weather and climate models (Wulfmeyer et al., 2016). In this work we present the high-resolution measurement results of the CBL and associated EZ using a recently-developed titled polarization lidar (TPL) over Wuhan (30.5°N, 114.4°E). The TPL is housed in a specially-customized working container and capable of operating under various weather conditions (including heavy precipitation). The TPL has an inclined working angle of 30° off zenith and routinely monitors the atmosphere with a time resolution of 10 s and a height resolution of 6.5 m. The equivalent minimum height with full overlap for the TPL is ~173 m above ground level (AGL). Based on the TPL-measured backscatter, a new approach has been developed for determination of the EZT. The small-scale characteristics of the CBL and associated EZ have also been investigated which can contribute to the improvement of understanding the structures and variations of the ABL, as well as parameterization of the EZ. The instrument,

methodology, observational results and summary and conclusions are stated succes-sively in following sections" to state the meaning, significance and novelty of this work. We feel that the introduction part has been greatly improved after modification.

2. Line 214: ABR should be defined when it shows up for the first time.

Authors' response: Along the Referee's suggestion, the ABR has been defined at an earlier place "In this work the variance profile of aerosol backscatter ratio (ABR) fluctu-ations is calculated and the height with maximum variance is assigned as ABL depth" (Please see Line 192 in the revised manuscript).

3. Line 242-246: Please give a figure to compare the ABR results.

Authors' response: Following the Referee's suggestion, we have provided a Figure S1 as an example in the Supplement to compare the ABR results obtained by the two lidars on January 31, 2020. From this comparison figure, it can be seen that the concurrent ABR results by these two lidars generally had nearly identical structures and comparable magnitudes in the ABL region. Stronger ABR determined by the TPL at the near range (<400 m), can be explained by different viewing geometry (30°-off for TPL and vertical for another PL) and the higher height ($\sim$300 m) of complete FOV for another PL.

Please also note the supplement to this comment:
https://acp.copernicus.org/preprints/acp-2020-963/acp-2020-963-AC2-supplement.pdf

---

## Author Comment (AC3) · 3 Jan 2021

General Comments: This manuscript is one part of an increasingly long list of papers simultaneously investigating the evolution of both planetary boundary layer (PBL) height and entrainment zone thickness for the haze events in China. Most have focused on CBL; this is one of the few to deal with EZ. It is of essence to investigate the variation of EZT near the PBL top, since it concerns the formation of cloud, the interaction of land-atmosphere, and the vertical mixing of scalars. The retrieval methods are scientifically robust, and the results interpretation makes sense, as far as I can tell. Therefore, I recommend acceptance for publication after addressing the following

concerns.

Authors' response: We greatly appreciate this Referee for the thoughtful considerations on the manuscript as well as encouraging and valuable comments to our work. In response to all the Referee's constructive suggestions, we have made necessary modifications point by point in the revised manuscript.

Major comments: 1. The title of this manuscript seems overstated. Actually, the authors only dealt with two cases from lidar measurements in Wuhan. Therefore, the title should be revised.

Authors' response: We thank the Referee for the pertinent comment on the title. Along the Referee's suggestion, now the renewed title reads "Characteristics of clear-day convective boundary layer and associated entrainment zone as observed by a ground-based polarization lidar over Wuhan (30.5°N, 114.4°E)" to state that the results are limited to clear-day weather conditions and to the observational location of Wuhan.

2. Most of the sentences are almost the same in both Conclusion and Abstract, especially regarding the statistic results of EZT evolution at different stages for both winter and summer cases. This should be avoided. The authors are suggested to highlight the major findings as well as the importance or implications of their work in Abstract, rather than simply duplicating the numbers.

Authors' response: We greatly appreciate the Referee for the valuable comments on the Abstract. Following the Referee's suggestions, the Abstract has now been modified as "Knowledge on the convective boundary layer (CBL) and associated entrainment zone (EZ) is significant for understanding the interaction of land-atmosphere and assessing the living conditions in the biosphere. A tilted 532-nm polarization lidar (30 degree off zenith) has been used for the routine atmospheric measurements with 10-s time and 6.5-m height resolution over Wuhan (30.5°N, 114.4°E). From lidar-retrieved aerosol backscatter, instantaneous ABL depths are obtained by logarithm gradient method and Harr wavelet transform method, while hourly-mean ABL depths

by variance method. A new approach utilizing the full width at half maximum of the variance profile of aerosol backscatter ratio fluctuations is proposed to determine the entrainment zone thickness (EZT). Four typical clear-day observational cases in different seasons are presented. The CBL evolution is described and studied in four (formation, growth, quasi-stationary and decay) developing stages; the instantaneous CBL depths exhibited different fluctuation magnitudes in the four stages and fluctuations at the growth stage were generally larger. The EZT is investigated for the same statistical time interval of 0900-1900 LT. It is found the winter and the late autumn cases had overall smaller mean (mean) and standard deviation (stddev) of EZT data than those of the late spring and early autumn cases. This statistical conclusion was also true for each of the four developing stages. Besides, compared to those of the late spring and early autumn cases, the winter and the late autumn cases had larger percentages of EZT falling into the subranges of 0-50 m but smaller percentages of EZT falling into the subranges of >150 m. It seems that both the EZT statistics (mean and stddev) and percentage of larger EZT value provide measures of entrainment intensity. Common statistical characteristics also existed. All four cases showed moderate variations of mean of EZT from stage to stage. The growth stage always had the largest mean and stddev of EZT and the quasi-stationary stage usually the smallest stddev of EZT. For all four stages, most EZT values fell into the 50-150 m subrange; the overall percentages of EZT falling into the 50-150 m subrange between 0900 and 1900 LT were >67% for all four cases. We believe that the lidar-derived characteristics of the clear-day CBL and associated EZ can contribute to improvement of understanding the structures and variations of the CBL, as well as providing quantitatively observational basis for EZ parameterization in numerical models" to highlight the major findings as well as their importance.

Minor comments: 1. L17: the first "FWHM" is redundant and can be deleted.

Authors' response: The first "FWHM" has already been removed in the revised manuscript.

2. L54: "despite of" -> "despite"

Authors' response: Now "despite of" has been changed to "despite" (Line 57 in the revised manuscript).

3. L77: "EZT" is a geophysical parameter rather than an approach. The authors mean "the determination of EZT"?

Authors' response: We thank the Referee for pointing out our inaccurate expression. Now the relevant sentence has been corrected to "The entrainment zone thickness (EZT) provides a possibility for parameterizing the entrainment rate" (Line 82 in the revised manuscript).

4. L181: "Jan" is not official acronym for "January", and should be given full spelling. All instances should be corrected throughout the MS.

Authors' response: Following the Referee's suggestion, now all "Jan" has been replaced by "January" in the revised manuscript.

5. L193: "measuring"-> "measurement"

Authors' response: We have changed "measuring" to "measurement" in the revised manuscript (Line 214 in the revised manuscript).

6. L245:"convinced"-> "confirmed"

Authors' response: Now "convinced" has been replaced by "confirmed" in the revised manuscript (Line 268 in the revised manuscript).

7. L432: "This new approach is designated as FWHM method in this work." can be deleted.

Authors' response: The sentence "This new approach is designated as FWHM method in this work." has now been removed in the revised manuscript.

Please also note the supplement to this comment:
https://acp.copernicus.org/preprints/acp-2020-963/acp-2020-963-AC3-supplement.pdf
* * *
* * *
[Figure]

**Supplement:**

*Supplement of*

**Measurement report: Characteristics of clear-day convective boundary layer and associated entrainment zone as observed by a ground-based polarization lidar over Wuhan (30.5 °N, 114.4 °E)**

**Fuchao Liu et al.**

*Correspondence to*: Fuchao Liu (lfc@whu.edu.cn), Fan Yi (yf@whu.edu.cn)

[Figure]

**Figure S1: Contour plots of ABR results obtained by (a) the TPL in this work and (b) another co-located vertically-pointing 532-nm PL (Kong and Yi, 2015) on January 31, 2020. The TPL has 10-s temporal and 6.5-m spatial resolution, while the PL has 60-s temporal and 30-m spatial resolution. The two lidars have a horizontal range of ~15 m apart. Note the concurrent ABR results by these two lidars generally had nearly identical structures and comparable magnitudes in the ABL region.**

[Figure]

**Figure S2: Same as Figure 4 but on the day of September 7, 2020.**

[Figure]

**Figure S3: Same as Figure 6 but on the day of September 7, 2020.**

**Table S1: Statistics of EZT obtained on September 7, 2020**

| Stage of CBL | | Formation | Growth | Quasi-stationary | Decay | Total |
|---|---|---|---|---|---|---|
| Time Interval (LT) | | 0900-1130 | 1130-1430 | 1430-1630 | 1630-1900 | 0900-1900 |
| Statistical data of EZT(km) | *min* | 0.039 | 0.039 | 0.041 | 0.034 | 0.034 |
| | *max* | 0.289 | 0.237 | 0.279 | 0.287 | 0.289 |
| | *mean* | 0.111 | 0.123 | 0.113 | 0.106 | 0.113 |
| | *stddev* | 0.058 | 0.062 | 0.057 | 0.060 | 0.060 |
| Percentages in each EZT subrange (%) | 0.00-0.05 km | 10.0 | 6.7 | 5.0 | 10.0 | 8.0 |
| | 0.05-0.10 km | 46.0 | 45.0 | 47.5 | 44.0 | 45.5 |
| | 0.10-0.15 km | 20.0 | 18.3 | 22.5 | 30.0 | 22.5 |
| | 0.15-0.20 km | 16.0 | 11.7 | 12.5 | 8.0 | 12.0 |
| | 0.20-0.30 km | 8.0 | 18.3 | 12.5 | 8.0 | 12.0 |

[Figure]

**Figure S4: Same as Figure 4 but on the day of November 12, 2020.**

[Figure]

**Figure S5: Same as Figure 6 but on the day of November 12, 2020.**

**Table S2: Statistics of EZT obtained on November 12, 2020**

| Stage of CBL | | Formation | Growth | Quasi-stationary | Decay | Total |
|---|---|---|---|---|---|---|
| Time Interval (LT) | | 0900-1130 | 1130-1430 | 1430-1630 | 1630-1900 | 0900-1900 |
| Statistical data of EZT(km) | *min* | 0.024 | 0.043 | 0.039 | 0.024 | 0.024 |
| | *max* | 0.182 | 0.257 | 0.244 | 0.182 | 0.257 |
| | *mean* | 0.084 | 0.133 | 0.106 | 0.092 | 0.103 |
| | *stddev* | 0.041 | 0.062 | 0.040 | 0.042 | 0.050 |
| Percentages in each EZT subrange (%) | 0.00-0.05 km | 22.0 | 5.0 | 5.0 | 14.0 | 11.5 |
| | 0.05-0.10 km | 48.0 | 30.0 | 48.3 | 46.0 | 44.0 |
| | 0.10-0.15 km | 22.0 | 22.5 | 28.3 | 32.0 | 26.5 |
| | 0.15-0.20 km | 8.0 | 22.5 | 13.4 | 8.0 | 12.5 |
| | 0.20-0.26 km | 0.0 | 20.0 | 5.0 | 0.0 | 5.5 |

**Table S3: Surface temperature magnitude ranges for each month and the four typical days in 2020 over Wuhan (30.5 °N, 114.4 °E)**

| Month | Surface temperature magnitude range |
|---|---|
| January | -1-13 ℃ (0-13 ℃ for January 31, 2020; **Case 1**) |
| February | 0-25 ℃ |
| March | 3-27 ℃ |
| April | 5-30 ℃ |
| May | 11-35 ℃ (18-29 ℃ for May 19, 2020; **Case 2**) |
| June | 21-34 ℃ |
| July | 21-34 ℃ |
| August | 20-37 ℃ |
| September | 15-34 ℃ (21-34 ℃ for September 7, 2020; **Case 3**) |
| October | 9-26 ℃ |
| November | 1-24 ℃ (7-22 ℃ for November 12, 2020; **Case 4**) |
| December | -6-16 ℃ |